# Preventing Gradient Attenuation in Lipschitz Constrained Convolutional Networks

**Qiyang Li**\*, **Saminul Haque**\*, **Cem Anil, James Lucas, Roger Grosse, Jörn-Henrik Jacobsen**
University of Toronto, Vector Institute
{qiyang.li, saminul.haque, cem.anil}@mail.utoronto.ca
{jlucas, rgrosse}@cs.toronto.edu
j.jacobsen@vectorinstitute.ai

## Abstract

Lipschitz constraints under $L_2$ norm on deep neural networks are useful for provable adversarial robustness bounds, stable training, and Wasserstein distance estimation. While heuristic approaches such as the gradient penalty have seen much practical success, it is challenging to achieve similar practical performance while provably enforcing a Lipschitz constraint. In principle, one can design Lipschitz constrained architectures using the composition property of Lipschitz functions, but Anil et al. [2] recently identified a key obstacle to this approach: gradient norm attenuation. They showed how to circumvent this problem in the case of fully connected networks by designing each layer to be gradient norm preserving. We extend their approach to train scalable, expressive, provably Lipschitz convolutional networks. In particular, we present the Block Convolution Orthogonal Parameterization (BCOP), an expressive parameterization of orthogonal convolution operations. We show that even though the space of orthogonal convolutions is disconnected, the largest connected component of BCOP with $2n$ channels can represent arbitrary BCOP convolutions over $n$ channels. Our BCOP parameterization allows us to train large convolutional networks with provable Lipschitz bounds. Empirically, we find that it is competitive with existing approaches to provable adversarial robustness and Wasserstein distance estimation. [2]

## 1 Introduction

There has been much interest in training neural networks with known upper bounds on their Lipschitz constants under $L_2$ norm[3]. Enforcing Lipschitz constraints can provide provable robustness against adversarial examples [48], improve generalization bounds [46], and enable Wasserstein distance estimation [2, 3, 22]. Heuristic methods for enforcing Lipschitz constraints, such as the gradient penalty [22] and spectral norm regularization [52], have seen much practical success, but provide no guarantees about the Lipschitz constant. It remains challenging to achieve similar practical success while provably satisfying a Lipschitz constraint.

In principle, one can design provably Lipschitz-constrained architectures by imposing a Lipschitz constraint on each layer; the Lipschitz bound for the network is then the product of the bounds for each layer. Anil et al. [2] identified a key difficulty with this approach: because a layer with a Lipschitz bound of 1 can only reduce the norm of the gradient during backpropagation, each step of backprop gradually attenuates the gradient norm, resulting in a much smaller Jacobian for the network's function than is theoretically allowed. We refer to this problem as *gradient norm attenuation*. They showed that Lipschitz-constrained ReLU networks were prevented from using

their full nonlinear capacity due to the need to prevent gradient norm attenuation. To counteract this problem, they introduced *gradient norm preserving (GNP)* architectures, where each layer preserves the gradient norm. For fully connected layers, this involved constraining the weight matrix to be orthogonal and using a GNP activation function called GroupSort. Unfortunately, the approach of Anil et al. [2] only applies to fully-connected networks, leaving open the question of how to constrain the Lipschitz constants of convolutional networks.

\*Equal contributions
[2]Code is available at: `github.com/ColinQiyangLi/LConvNet`
[3]Unless specified otherwise, we refer to Lipschitz constant as the Lipschitz constant under $L_2$ norm.

As many state-of-the-art deep learning applications rely on convolutional networks, there have been numerous attempts for tightly enforcing Lipschitz constants of convolutional networks. However, those existing techniques either hinder representational power or induce difficulty in optimization. Cisse et al. [12], Tsuzuku et al. [48], Qian and Wegman [40] provide loose bounds on the Lipschitz constant that can limit the parameterizable region. Gouk et al. [21] obtain a tight bound on the Lipschitz constant, but tend to lose expressive power during training due to vanishing singular values. The approach of Sedghi et al. [45] is computationally intractable for larger networks.

In this work, we introduce convolutional GNP networks with an efficient parameterization of orthogonal convolutions by adapting the construction algorithm from Xiao et al. [51]. This parameterization avoids the issues of loose bounds on the Lipschitz constant and computational intractability observed in the aforementioned approaches. Furthermore, we provide theoretical analysis that demonstrates the disconnectedness of the orthogonal convolution space, and how our parameterization alleviates the optimization challenge engendered by the disconnectedness.

We evaluate our GNP networks in two situations where expressive Lipschitz-constrained networks are of central importance. The first is provable norm-bounded adversarial robustness, which is the task of classification and additionally certifying that the network's classification will not change under any norm-bounded perturbation. Due to the tight Lipschitz properties, the constructed GNP networks can easily give non-trivial lower bounds on the robustness of the network's classification. We demonstrate that our method outperforms the state-of-the-art in provable deterministic robustness under $L_2$ metric on MNIST and CIFAR-10. The other application is Wasserstein distance estimation. Wasserstein distance estimation can be rephrased as a maximization over 1-Lipschitz functions, allowing our Lipschitz-constrained networks to be directly applied to this problem. Moreover, the restriction to GNP we impose is not necessarily a hindrance, as it is shown by Gemici et al. [19] that the optimal 1-Lipschitz function is also GNP almost everywhere. We demonstrate that our GNP convolutional networks can obtain tighter Wasserstein distance estimates than competing architectures.

## 2 Background

### 2.1 Lipschitz Functions under $L_2$ Norm

In this work, we focus on Lipschitz functions with respect to the $L_2$ norm. We say a function $f : \mathbb{R}^n \to \mathbb{R}^m$ is $l$-Lipschitz if and only if

$$||f(\mathbf{x}_1) - f(\mathbf{x}_2)||_2 \leq l||\mathbf{x}_1 - \mathbf{x}_2||_2, \forall \mathbf{x}_1, \mathbf{x}_2 \in \mathbb{R}^n \tag{1}$$

We denote $\mathrm{Lip}(f)$ as the smallest $K$ for which $f$ is $l$-Lipschitz, and call it the **Lipschitz constant** of $f$. For two Lipschitz continuous functions $f$ and $g$, the following property holds:

$$\mathrm{Lip}(f \circ g) \leq \mathrm{Lip}(f)\,\mathrm{Lip}(g) \tag{2}$$

The most basic neural network design consists of a composition of linear transformations and non-linear activation functions. The property above (Equation 2) allows one to upper-bound the Lipschitz constant of a network by the product of the Lipschitz constants of each layer. However, as modern neural networks tend to possess many layers, the resultant upper-bound is likely to be very loose, and constraining it increases the risk of diminishing the Lipschitz constrained network capacity that can be utilized.

### 2.2 Gradient Norm Preservation (GNP)

Let $\mathbf{y} = f(\mathbf{x})$ be 1-Lipschitz, and $\mathcal{L}$ be a loss function. The norm of the gradient after backpropagating through a 1-Lipschitz function is no larger than the norm of the gradient before doing so:

$$\|\nabla_{\mathbf{x}}\mathcal{L}\|_2 = \|(\nabla_{\mathbf{y}}\mathcal{L})(\nabla_{\mathbf{x}}f)\|_2 \leq \|\nabla_{\mathbf{y}}\mathcal{L}\|_2 \|\nabla_{\mathbf{x}}f\|_2 \leq \mathrm{Lip}(f) \|\nabla_{\mathbf{y}}\mathcal{L}\|_2 \leq \|\nabla_{\mathbf{y}}\mathcal{L}\|_2$$

As a consequence of this relation, the gradient norm will likely be attenuated during backprop if no special measures are taken. One way to fix the gradient norm attenuation problem is to enforce

each layer to be gradient norm preserving (GNP). Formally, $f : \mathbb{R}^n \mapsto \mathbb{R}^m$ is GNP if and only if its input-output Jacobian, $J \in \mathbb{R}^{m \times n}$, satisfies the following property:

$$\left\| J^T \mathbf{g} \right\|_2 = ||\mathbf{g}||_2, \forall \mathbf{g} \in \mathcal{G}.$$

where $\mathcal{G} \subseteq \mathbb{R}^m$ defines the possible values that the gradient vector $\mathbf{g}$ can take. Note that when $m = n$, this condition is equivalent to orthogonality of $J$. In this work, we consider a slightly stricter definition where $\mathcal{G} = \mathbb{R}^m$ because this allows us to directly compose two GNP (strict) functions without reasoning about their corresponding $\mathcal{G}$. For the rest of the paper, unless specified otherwise, a GNP function refers to this more strict definition.

Based on the definition of GNP, we can deduce that GNP functions are 1-Lipschitz in the 2-norm. Since the composition of GNP functions is also GNP, one can design a GNP network by stacking GNP building blocks. Another favourable condition that GNP networks exhibit is *dynamical isometry* [51, 37, 38] (where the entire distribution of singular values of input-output Jacobian is close to 1), which has been shown to improve training speed and stability.

### 2.3 Provable Norm-bounded Adversarial Robustness

We consider a classifier $f$ with $T$ classes that takes in input $\mathbf{x}$ and produces a logit for each of the classes: $f(\mathbf{x}) = [y_1 \quad y_2 \quad \cdots \quad y_T]$. An input data point $\mathbf{x}$ with label $t \in \{1, 2, \cdots, T\}$ is provably robustly classified by $f$ under perturbation norm of $\epsilon$ if

$$\arg \max_i f(\mathbf{x} + \boldsymbol{\delta})_i = t, \forall \boldsymbol{\delta} : ||\boldsymbol{\delta}||_2 \leq \epsilon.$$

The margin of the prediction for $\mathbf{x}$ is given by $\mathcal{M}_f(\mathbf{x}) = \max(0, y_t - \max_{i \neq t} y_i)$. If $f$ is $l$-Lipschitz, we can certify that $f$ is robust with respect to $\mathbf{x}$ if $\sqrt{2} l \epsilon < \mathcal{M}_f(\mathbf{x})$ (See Appendix P for the proof).

### 2.4 Wasserstein Distance Estimation

Wasserstein distance is a distance metric between two probability distributions [39]. The Kantorovich-Rubinstein formulation of Wasserstein distance expresses it as a maximization problem over 1-Lipschitz functions [3]:

$$W(P_1, P_2) = \sup_{f : \text{Lip}(f) \leq 1} \left( \mathbb{E}_{\mathbf{x} \sim P_1(\mathbf{x})}[f(\mathbf{x})] - \mathbb{E}_{\mathbf{x} \sim P_2(\mathbf{x})}[f(\mathbf{x})] \right). \tag{3}$$

In Wasserstein GAN architecture, Arjovsky et al. [3] proposed to parametrize the scalar-valued function $f$ using a Lipschitz constrained network, which serves as the discriminator that estimates the Wasserstein distance between the generator and data distribution. One important property to note is that the optimal scalar function $f$ is GNP almost everywhere (See Corollary 1 in Gemici et al. [19]). Naturally, this property favours the optimization approach that focuses on searching over GNP functions. Indeed, Anil et al. [2] found that GNP networks can achieve tighter lower bounds compared to non-GNP networks.

## 3 Orthogonal Convolution Kernels

The most crucial step to building a GNP convolutional network is constructing the GNP convolution itself. Since a convolution operator is a linear operator, making the convolution kernel GNP is equivalent to making its corresponding linear operator orthogonal. While there are numerous methods for orthogonalizing arbitrary linear operators, it is not immediately clear how to do this for convolutions, especially when preserving kernel size. We first summarize the orthogonal convolution representations from Kautsky and Turcajová [28] and Xiao et al. [51] (Section 3.1). Then, we analyze the topology of the space of orthogonal convolution kernels and demonstrate that the space is disconnected (with at least $\mathcal{O}(n^2)$ connected components for a $2 \times 2$ 2-D convolution layer), which is problematic for gradient-based optimization methods because they are confined to one component (Section 3.2). Fortunately, this problem can be fixed by increasing the number of channels: we demonstrate that a single connected component of the space of orthogonal convolutions with $2n$ channels can represent any orthogonal convolution with $n$ channels (Section 3.3).

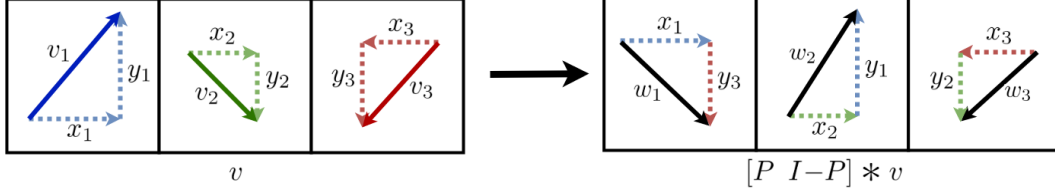

Figure 1: Visualization of a 1-D orthogonal convolution, $[P \ \ I - P]$, applied to a 1-D input tensor $v \in \mathbb{R}^{2 \times 3}$ with a length of 3 and channel size of 2. $P \in \mathbb{R}^{2 \times 2}$ here is the orthogonal projection onto the $x$-axis, which makes $I - P$ the complementary projection onto the $y$-axis. Each cell of $v$ corresponds to one of the three spatial locations, and the vector contained within it represents the vector along the channel dimension in said spatial location.

### 3.1 Constructing Orthogonal Convolutions

To begin analysing orthogonal convolution kernels, we must first understand the symmetric projector, which is a fundamental building block of orthogonal convolutions. An $n \times n$ matrix $P$ is defined to be a symmetric projector if $P = P^2 = P^T$. Geometrically, a symmetric projector $P$ represents an orthogonal projection onto the range of $P$. From this geometric interpretation, it is not hard to see that the space of projectors has $n + 1$ connected components, based on the rank of the projector (for a more rigorous treatment, see Remark 4.1 in Appendix K). For notation simplicity, we denote $\mathbb{P}(n)$ as the set of all $n \times n$ symmetric projectors and $\mathbb{P}(n, k)$ as the subset of all $n \times n$ symmetric projectors with ranks of $k$.

Now that the concept of symmetric projectors has been established, we will consider how to construct 1-D convolutional kernels. As shown by Kautsky and Turcajová [28], all 1-D orthogonal convolution kernels with a kernel size $K$ can be represented as:

$$\mathcal{W}(H, P_{1:K-1}) = H \square [P_1 \quad (I - P_1)] \square \cdots \square [P_{K-1} \quad (I - P_{K-1})] \tag{4}$$

where $H \in O(n)$ is an orthogonal matrix of $n \times n$, $P_i \in \mathbb{P}(n)$, and $\square$ represents block convolution, which is convolution using matrix multiplication rather than scalar multiplication:

$$[X_1 \quad X_2 \quad \cdots \quad X_p] \square [Y_1 \quad Y_2 \quad \cdots \quad Y_q] = [Z_1 \quad Z_2 \quad \cdots \quad Z_{p+q-1}]$$

with $Z_i = \sum_{i'=-\infty}^{\infty} X_{i'} Y_{i-i'}$, where the out-of-range elements are all zero (e.g., $X_{<1} = 0, X_{>p} = 0, Y_{<1} = 0, Y_{>q} = 0$). Unlike regular convolutions, the block convolution does not commute since matrix multiplication does not commute. One important property of block convolution is that it corresponds to composition of the kernel operators. That is, $X * (Y * v) = (X \square Y) * v$, where $A * v$ represents the resulting tensor after applying convolution $A$ to $v$. This composition property allows us to decompose the representation (Equation 4) into applications of orthogonal convolutions with kernel size of 2 (Figure 1 demonstrates the effect of it) along with a channel-wise orthogonal transformation ($H$).

Xiao et al. [51] extended the 1-D representation to the 2-D case using alternating applications of orthogonal convolutions of size 2:

$$\mathcal{W}(H, P_{1:K-1}, Q_{1:K-1}) = H \square \begin{bmatrix} P_1 \\ I - P_1 \end{bmatrix} \square [Q_1 \quad I - Q_1] \square \cdots$$
$$\cdots \square \begin{bmatrix} P_{K-1} \\ I - P_{K-1} \end{bmatrix} \square [Q_{K-1} \quad I - Q_{K-1}] \tag{5}$$

where $Z = X \square Y$ is defined similarly to the 1-D case with $Z_{ij} = \sum_{i'=-\infty}^{\infty} \sum_{j'=-\infty}^{\infty} [X_{i',j'} Y_{i-i',j-j'}]$, and $P_i, Q_i \in \mathbb{P}(n)$. Unlike in 1-D, we discovered that this 2-D representation could only represent a subset of 2-D orthogonal convolutions (see Appendix O for an example). However, we do not know whether simple modifications to this parameterization will result in a complete representation of all 2-D orthogonal convolutions (see Appendix O for details on the open question).

### 3.2 Topology of the Orthogonal Convolution Space

Before utilizing this space of orthogonal convolutions, we would like to analyze some fundamental properties of this space. Since $\mathbb{P}(n)$ has $n + 1$ connected components and orthogonal convolutions

**Algorithm 1:** Block Convolution Orthogonal Parameterization (BCOP)

---

**Input:** $c_o \times c_i$ unconstrained matrix $O$, $c_i \times \lfloor \frac{c_i}{2} \rfloor$ unconstrained matrices $M_i, N_i$ for $i$ from 1 to $k-1$, assuming $c_i \geq c_o$

**Result:** Orthogonal Convolution Kernel $W \in \mathbb{R}^{c_o \times c_i \times K \times K}$

$H \leftarrow \text{Orthogonalize}(O)$; ▷ any differentiable orthogonalization procedure (e.g., Björck [5]);

Initialize $W$ as a $1 \times 1$ convolution with $W[0,0] = H$;

**for** *i from 1 to $K-1$* **do**

$\quad R_P, R_Q \leftarrow \text{Orthogonalize}(M_i), \text{Orthogonalize}(N_i)$;

$\quad P, Q \leftarrow R_P R_P^T, R_Q R_Q^T$; ▷ Construct symmetric projectors with half of the full rank;

$\quad W \leftarrow W \square \begin{bmatrix} P \\ I - P \end{bmatrix} \square \begin{bmatrix} Q & I - Q \end{bmatrix}$

**end**

**Output:** $W$

---

are constructed out of many projectors, it is to be expected that there are many connected components in the space of orthogonal convolutions. Indeed, we see the first result in 1-D (Theorem 1).

**Theorem 1** (Connected Components of 1-D Orthogonal Convolution)**.** *The 1-D orthogonal convolution space is compact and has $2(K-1)n + 2$ connected components, where $K$ is the kernel size and $n$ is the number of channels.*

In 2-D, we analyze case of kernel size of 2 ($2 \times 2$ kernels) and show that the number of connected components grows at least quadratically with respect to the channel size:

**Theorem 2** (Connected Components of 2-D Orthogonal Convolution with $K = 2$)**.** *2-D orthogonal convolution space with a kernel size of $2 \times 2$ has at least $2(n+1)^2$ connected components, where $n$ is the number of channels.*

The disconnectedness in the space of orthogonal convolution imposes an intrinsic difficulty in optimizing over the space of orthogonal convolution kernels, as gradient-based optimizers are confined to their initial connected component. We refer readers to Appendix K for the proof of Theorem 1 and Appendix M for the proof of Theorem 2.

### 3.3   Block Convolution Orthogonal Parameterization (BCOP)

To remedy the disconnectedness issue, we show the following:

**Theorem 3** (BCOP Construction with Auxiliary Dimension)**.** *For any convolution $C = \mathcal{W}(H, P_{1:K-1}, Q_{1:K-1})$ with input and output channels $n$ and $P_i, Q_i \in \mathbb{P}(n)$, there exists a convolution $C' = \mathcal{W}(H', P'_{1:K-1}, Q'_{1:K-1})$ with input and output channels $2n$ constructed from only $n$-rank projectors ($P'_i, Q'_i \in \mathbb{P}(2n, n)$) such that $C'(\mathbf{x})_{1:n} = C(\mathbf{x}_{1:n})$. That is, the first $n$ channels of the output is the same with respect to the first $n$ channels of the input under both convolutions.*

The idea behind this result is that some projectors in $\mathbb{P}(2n, n)$ may use their first $n$ dimensions to represent $\mathbb{P}(n)$ and then use the latter $n$ dimensions in a trivial capacity so that the total rank is $n$ (see Appendix N for the detailed proof).

Theorem 3 implies that all connected components of orthogonal convolutions constructed by $\mathcal{W}$ with $n$ channels can all be equivalently represented in a single connected component of convolutions constructed by $\mathcal{W}$ with $2n$ channels by only using projectors that have rank $n$. (This comes at the cost of requiring 4 times as many parameters.)

This result motivates us to parameterize the connected subspace of orthogonal convolutions defined by $\mathcal{W}(H, \tilde{P}_{1:K-1}, \tilde{Q}_{1:K-1})$ where $\tilde{P}_i \in \mathbb{P}(n, \lfloor \frac{n}{2} \rfloor)$ and $\tilde{Q}_i \in \mathbb{P}(n, \lfloor \frac{n}{2} \rfloor)$. We refer to this method as the Block Convolution Orthogonal Parameterization (BCOP). The procedure for BCOP is summarized in Algorithm 1 (See Appendix H for implementation details).

## 4   Related Work

**Reshaped Kernel Method (RK)**   This method reshapes a convolution kernel with dimensions $(c_o, c_i, k, k)$ into a $(c_o, k^2 c_i)$ matrix. The Lipschitz constant (or spectral norm) of a convolution

operator is bounded by a constant factor of the spectral norm of its reshaped matrix [12, 48, 40], which enables bounding of the convolution operator's Lipschitz constant by bounding that of the reshaped matrix. However, this upper-bound can be conservative, causing a bias towards convolution operators with low Lipschitz constants, limiting the method's expressive power. In this work, we strictly enforce orthogonality of the reshaped matrix rather than softly constrain it via regularization, as done in Cisse et al. [12]. We refer to this variant as **reshaped kernel orthogonalization**, or **RKO**.

**One-Sided Spectral Normalization (OSSN)**   This variant of spectral normalization [36] scales the kernel so that the spectral norm of the convolution operator is at most 1 [21]. This is a projection under the matrix 2-norm but *not* the Frobenius norm. It is notable because when using Euclidean steepest descent with this projection, such as in constrained gradient-based optimization, there is no guarantee to converge to the correct solution (see an explicit example and further analysis in Appendix A). In practice, we found that projecting during the forward pass (as in Miyato et al. [36]) yields better performance than projecting after each gradient update.

**Singular Value Clipping and Masking (SVCM)**   Unlike spectral normalization, singular value clipping is a valid projection under the Frobenius norm. Sedghi et al. [45] demonstrates a method to perform an approximation of the optimal projection to the orthogonal kernel space. Unfortunately, this method needs many expensive iterations to enforce the Lipschitz constraint tightly, making this approach computationally intractable in training large networks with *provable* Lipschitz constraints.

**Comparison to BCOP**   OSSN and SVCM's run-time depend on the input's spatial dimensions, which prohibits scalability (See Appendix C for a time complexity analysis). RKO does not guarantee an exact Lipschitz constant, which may cause a loss in expressive power. Additionally, none of these methods guarantee gradient norm preservation. BCOP avoids all of the issues above.

## 4.1   Provable Adversarial Robustness

Certifying the adversarial robustness of a network subject to norm ball perturbation is difficult. Exact certification methods using mixed-integer linear programming or SMT solvers scale poorly with the complexity of the network [27, 10]. Cohen et al. [15] and Salman et al. [43] use an estimated smoothed classifier to achieve very high provable robustness with high confidence. In this work, we are primarily interested in providing deterministic provable robustness guarantees.

Recent work has been focusing on guiding the training of the network to be verified or certified (providing a lower-bound on provable robustness) easier [49, 50, 18, 17, 23]. For example, Xiao et al. [50] encourage weight sparsity and perform network pruning to speed up the exact verification process for ReLU networks. Wong et al. [49] optimize the network directly towards a robustness lower-bound using a dual optimization formulation.

Alternatively, rather than modifying the optimization objective to incentivize robust classification, one can train networks to have a small global Lipschitz constant, which allows an easy way to certify robustness via the output margin. Cohen et al. [14] deploy spectral norm regularization on weight matrices of a fully connected network to constrain the Lipschitz constant and certify the robustness of the network at the test time. Tsuzuku et al. [48] estimate an upper-bound of the network's Lipschitz constant and train the network to maximize the output margin using a modified softmax objective function according to the estimated Lipschitz constant. In contrast to these approaches, Anil et al. [2] train fully connected networks that have a known Lipschitz constant by enforcing gradient norm preservation. Our work extends this idea to convolutional networks.

## 5   Experiments

The primary point of interest for the BCOP method (Section 3.3) is its expressiveness compared against other common approaches of paramterizing Lipschitz constrained convolutions (Section 4). To study this, we perform an ablation study on two tasks using these architectures: The first task is provably robust image classification tasks on two datasets (MNIST [31] and CIFAR-10 [30])[4]. We find our method outperformed other Lipschitz constrained convolutions under the same architectures as well as the state-of-the-art in this task (Section 5.2). The second task is 1-Wasserstein distance

estimation of GANs where our method also outperformed other competing Lipschitz-convolutions under the same architecutre (Section 5.3).

## 5.1 Network Architectures and Training Details

A benefit of training GNP networks is that we enjoy the property of dynamical isometry, which inherently affords greater training stability, thereby reducing the need for common techniques that would otherwise be difficult to incorporate into a GNP network. For example, if a 1-Lipschitz residual connection maintains GNP, the residual block must be an identity function with a constant bias (see an informal justification in Appendix D.1). Also, batch normalization involves scaling the layer's output, which is not necessarily 1-Lipschitz, let alone GNP. For these reasons, residual connections and batch normalization are not included in the model architecture. We also use cyclic padding to substitute zero-padding since zero-padded orthogonal convolutions must be size 1 (see an informal proof in Appendix D.2). Finally, we use "invertible downsampling" [25] in replacement of striding and pooling to achieve spatial downsampling while maintaining the GNP property. The details for these architectural decisions are in Appendix D.

Because of these architectural constraints, we base our networks on architectures that do not involve residual connections. For provable robustness experiments, we use the "Small" and "Large" convolutional networks from Wong et al. [49]. For Wasserstein distance estimation, we use the fully convolutional critic from Radford et al. [41] (See Appendix E, F for details). Unless specified otherwise, each experiment is repeated 5 times with mean and standard deviation reported.

## 5.2 Provable Adversarial Robustness

### 5.2.1 Robustness Evaluation

For adversarial robustness evaluation, we use the $L_2$-norm-constrained threat model [8], where the adversary is constrained to $L_2$-bounded perturbation with the $L_2$ norm constrained to be below $\epsilon$. We refer to *clean accuracy* as the percentage of un-perturbed examples that are correctly classified and *robust accuracy* as the percentage of examples that are guaranteed to be correctly classified under the threat model. We use the margin of the model prediction to determine a lower bound of the robust accuracy (as described in Section 2.3). We also evaluate the empirical robustness of our model around under two gradient-based attacks and two decision-based attacks: *(i)* PGD attack with CW loss [34, 7], *(ii)* FGSM [47], *(iii)* Boundary attack (BA) [6], *(iv)* Point-wise attack (PA) [44]. Specifically, the gradient-based methods (*(i)* and *(ii)*) are done on the whole test dataset; the decision-based attacks (*(iii)* and *(iv)*) are done only on the first 100 test data points since they are expensive to run.[5]

### 5.2.2 Comparison of Different Methods for Enforcing Spectral Norm of Convolution

We compare the performance of OSSN, RKO, SVCM, and BCOP on margin training for adversarial robustness on MNIST and CIFAR-10. To make the comparison fair, we ensure all the methods have a tight Lipschitz constraint of 1. For OSSN, we use 10 power iterations and keep a running vector for each convolution layer to estimate the spectral norm and perform the projection during every forward pass. For SVCM, we perform the singular value clipping projection with 50 iterations after every 100 gradient updates to ensure the Lipschitz bound is tight. For RKO, instead of using a regularization term to enforce the orthogonality (as done in Cisse et al. [12]), we use Björck [5] to orthogonalize the reshaped matrix before scaling down the kernel. We train two different convolutional architectures with the four aforementioned methods of enforcing Lipschitz convolution layers on image classification tasks. To achieve large output margins, we use first-order, multi-class, hinge loss with a margin of 2.12 on MNIST and 0.7071 on CIFAR-10.

Our approach (BCOP) outperforms all competing methods across all architectures on both MNIST and CIFAR-10 (See Table 1 and Appendix I, Table 7). To understand the performance gap, we visualize the singular value distribution of a convolution layer before and after training in Figure 2. We observe that OSSN and RKO push many singular values to 0, suggesting that the convolution layer is not fully utilizing the expressive power it is capable of. This observation is consistent with our hypothesis that these methods bias the convolution operators towards sub-optimal regions caused by the loose Lipschitz bound (for RKO) and improper projection (for OSSN). In contrast, SVCM's singular values started mostly near 0.5 and some of them were pushed up towards 1, which is consistent with the

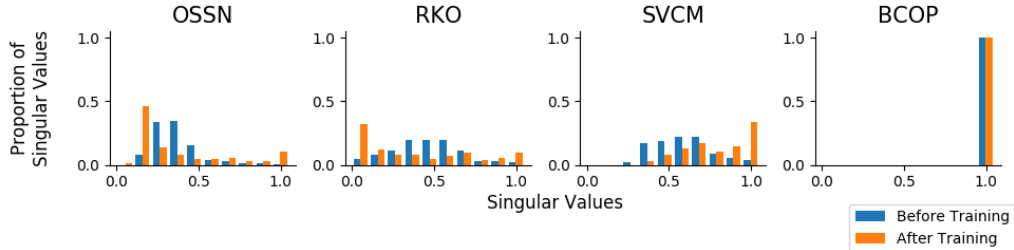

Figure 2: Singular value distribution at initialization (blue) and at the end of training (orange) for the second layer of the "Large" baseline using different methods to enforce Lipschitz convolution.

| Dataset | | | OSSN | RKO | SVCM | BCOP |
|---|---|---|---|---|---|---|
| **MNIST** ($\epsilon = 1.58$) | Small | Clean | $96.86 \pm 0.13$ | $97.28 \pm 0.08$ | $97.24 \pm 0.09$ | $\mathbf{97.54} \pm 0.06$ |
| | | Robust | $42.95 \pm 1.09$ | $43.58 \pm 0.44$ | $28.94 \pm 1.58$ | $\mathbf{45.84} \pm 0.90$ |
| | Large | Clean | $98.31 \pm 0.03$ | $98.44 \pm 0.05$ | $97.93 \pm 0.05$ | $\mathbf{98.77} \pm 0.05$ |
| | | Robust | $53.77 \pm 1.02$ | $55.18 \pm 0.46$ | $38.00 \pm 1.82$ | $\mathbf{56.66} \pm 0.23$ |
| **CIFAR-10** ($\epsilon = 36/255$) | Small | Clean | $62.18 \pm 0.66$ | $61.77 \pm 0.63$ | $62.39 \pm 0.46$ | $\mathbf{64.53} \pm 0.30$ |
| | | Robust | $48.03 \pm 0.54$ | $47.46 \pm 0.53$ | $47.59 \pm 0.56$ | $\mathbf{50.01} \pm 0.21$ |
| | Large | Clean | $67.51 \pm 0.47$ | $70.01 \pm 0.26$ | $69.65 \pm 0.38$ | $\mathbf{72.41} \pm 0.22$ |
| | | Robust | $53.64 \pm 0.49$ | $55.76 \pm 0.16$ | $53.61 \pm 0.51$ | $\mathbf{58.72} \pm 0.23$ |

Table 1: Clean and robust accuracy on MNIST and CIFAR-10 using different Lipschitz convolutions. The provable robust accuracy is evaluated at $\epsilon = 1.58$ for MNIST and at $\epsilon = 36/255$ for CIFAR-10.

| Dataset | | BCOP-Large | FC-3 | KW-Large | KW-Resnet |
|---|---|---|---|---|---|
| **MNIST** ($\epsilon = 1.58$) | Clean | $\mathbf{98.77} \pm 0.05$ | $98.71 \pm 0.02$ | $88.12$ | - |
| | Robust | $\mathbf{56.66} \pm 0.23$ | $54.46 \pm 0.30$ | $44.53$ | - |
| **CIFAR-10** ($\epsilon = 36/255$) | Clean | $\mathbf{72.41} \pm 0.22$ | $62.60 \pm 0.39$ | $59.76$ | $61.20$ |
| | Robust | $\mathbf{58.72} \pm 0.23$ | $49.97 \pm 0.35$ | $50.60$ | $51.96$ |

Table 2: Comparison of our convolutional networks and the fully connected baseline in Anil et al. [2] (FC-3) against provably robust models in previous works. The numbers for KW-Large and KW-Resnet are directly obtained from Table 4 of in the Appendix of their paper [49].

procedure being an optimal projection. BCOP has all of its singular values at 1 throughout training by design due to its gradient norm preservation and orthogonality. Thus, we empirically verify the downsides of other methods and show that our proposed method enables maximally expressive Lipschitz constrained convolutional layers with guaranteed gradient-norm-preservation.

### 5.2.3 State-of-the-art Comparison

To further demonstrate the expressive power of orthogonal convolution, we compare our networks with models that achieve state-of-the-art deterministic provable adversarial robustness performance (Table 2 and Appendix J, Table 8 and 9). We also evaluate the empirical robustness of our model against common attacks on CIFAR-10. Comparing against Wong et al. [49], our approach reaches similar performance for "Small" architecture and better performance for "Large" architecture (Table 3).

### 5.3 Wasserstein Distance Estimation

In this section, we consider the problem of estimating the Wasserstein distance between two high dimensional distributions using neural networks. Anil et al. [2] showed that in the fully connected setting, ensuring gradient norm preservation is critical for obtaining tighter lower bounds on the Wasserstein distance. We observe the same phenomenon in the convolutional setting.

|  | KW | BCOP |  |  | KW | BCOP |
|---|---|---|---|---|---|---|
| **Small** |  |  |  | **Small** |  |  |
| Clean | 54.39 | $\mathbf{64.53} \pm 0.30$ |  | Clean (*) | 63.00 | $\mathbf{74.20} \pm 2.23$ |
| PGD | 49.94 | $\mathbf{51.26} \pm 0.17$ |  | BA (*) | 60.00 | $\mathbf{61.20} \pm 2.99$ |
| FGSM | 49.98 | $\mathbf{51.57} \pm 0.18$ |  | PA (*) | 63.00 | $\mathbf{74.00} \pm 2.28$ |
| **Large** |  |  |  | **Large** |  |  |
| Clean | 60.14 | $\mathbf{72.41} \pm 0.22$ |  | Clean (*) | 68.00 | $\mathbf{77.60} \pm 1.74$ |
| PGD | 55.53 | $\mathbf{64.39} \pm 0.26$ |  | BA (*) | 64.00 | $\mathbf{71.20} \pm 1.60$ |
| FGSM | 55.55 | $\mathbf{64.53} \pm 0.25$ |  | PA (*) | 68.00 | $\mathbf{77.20} \pm 1.60$ |

Table 3: Comparison of our networks with Wong et al. [49] on CIFAR-10 dataset. Left: results of the evaluation on the entire CIFAR-10 test dataset. Right (*): results of the evaluation on the first 100 test samples. The KW models [49] are directly taken from their official repository.

|  |  | OSSN | RKO | BCOP |
|---|---|---|---|---|
| **STL-10** | **MaxMin** | $7.39 \pm 0.31$ | $8.95 \pm 0.12$ | $\mathbf{9.91} \pm 0.11$ |
|  | **ReLU** | $7.06 \pm 0.72$ | $7.82 \pm 0.21$ | $\mathbf{8.28} \pm 0.19$ |
| **CIFAR-10** | **MaxMin** | $3.29 \pm 0.05$ | $4.95 \pm 0.08$ | $\mathbf{5.34} \pm 0.07$ |
|  | **ReLU** | $3.07 \pm 0.12$ | $4.20 \pm 0.06$ | $\mathbf{4.39} \pm 0.07$ |

Table 4: Comparison of different Lipschitz constrained architectures on the Wasserstein distance estimation task between the data and generator distributions of STL-10 and CIFAR-10 GANs. Each estimate is a strict lower bound (estimated using 6,400 pairs of randomly sampled real and generated image examples), hence larger values indicate better performance.

We trained our networks to estimate the Wasserstein distance between the data and generator distributions of GANs[6] [20] trained on RGB images from the STL-10 dataset [13] and CIFAR-10 dataset [30] (resized to 64x64). After training the GANs, we froze the generator weights and trained Lipschitz constrained convolutional networks to estimate the Wasserstein distance. We adapted the fully convolutional discriminator model used by Radford et al. [41] by removing all batch normalization layers and replacing all vanilla convolutional layers with Lipschitz candidates (BCOP, RKO, and OSSN)[7]. We trained each model with ReLU or MaxMin activations [2]. The results are in Table 4.

Baking in gradient norm preservation in the architecture leads to significantly tighter lower bounds on the Wasserstein distance. The only architecture that has gradient norm preserving layers throughout (BCOP with MaxMin) leads to the best estimate. Although OSSN has the freedom to learn orthogonal kernels, this does not happen in practice and leads to poorer expressive power.

## 6 Conclusion and Future Work

We introduced convolutional GNP networks with an efficient construction method of orthogonal convolutions (BCOP) that overcomes the common issues of Lipschitz constrained networks such as loose Lipschitz bounds and gradient norm attenuation. In addition, we showed the space of orthogonal convolutions has many connected components and demonstrated how BCOP parameterization alleviates the optimization challenges caused by the disconnectedness. Our GNP networks outperformed the state-of-the-art for deterministic provable adversarial robustness on $L_2$ metrics with both CIFAR-10 and MNIST, and obtained tighter Wasserstein distance estimates between high dimensional distributions than competing approaches. Despite its effectiveness, our parameterization is limited to only expressing a subspace of orthogonal convolutions. A complete parameterization of the orthogonal convolution space may enable training even more powerful GNP convolutional networks. We presented potential directions to achieve this and left the problem for future work.

**Acknowledgements**

We would like to thank Lechao Xiao, Arthur Rabinovich, Matt Koster, and Siqi Zhou for their valuable insights and feedback. We would like to thank Sherjil Ozair for spotting a bug in our code, whose fix improved our results. RG acknowledges support from the CIFAR Canadian AI Chairs program.

## Footnotes

[2]Code is available at: `github.com/ColinQiyangLi/LConvNet`

[3]Unless specified otherwise, we refer to Lipschitz constant as the Lipschitz constant under $L_2$ norm.

[4]We only claim in the deterministic case as recent approaches have much higher probabilistic provable robustness [15, 43].

[5]We use foolbox [42] for the two decision-based methods.

[6]Note that any GAN variant could have been chosen here.

[7]We omit SVCM for comparison due to its computational intractability

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
