[Supplementary Material · Lipschitz_ConvNets_CameraReady_Final-Supplementary_2-22.pdf]

# Supplementary Materials for Preventing Gradient Attenuation in Lipschitz Constrained Convolutional Networks

## A    Optimizing under spectral normalization

Here we provide theoretical analysis of the optimization properties of spectral normalization. We focus on the setting in which the weight matrices are projected to the feasible set via spectral normalization after each gradient update (i.e. projected gradient descent).

Firstly, we note that spectral normalization is a valid projection under the operator 2-norm [10] but not the Frobenius norm, where the projection would clip all singular values larger than 1 [17]. Despite this, all existing implementations of spectral normalization as a projection perform steepest descent optimization in Euclidean space which is not guaranteed to converge [4]. We illustrate this with a simple example.

**Spectral norm projection counter-example**    Consider a constraint optimization problem:

$$A^* = \underset{A:||A||_2 \leq 1}{\arg\max} \{\operatorname{Tr}(AD)\}, \tag{1}$$

where $A$ and $D$ are diagonal, with $\operatorname{diag}(D) = [2, 1]$. Clearly, the objective is maximized by $A^* = I$. However, the Euclidean steepest ascent direction is given by the gradient, which is $D$ in this case. A single gradient update (with learning rate $\alpha$) and projection step acting on the diagonal of A looks like this (assuming that $x + 2\alpha > y + \alpha$ throughout the course of learning, which is indeed the case given the initialization):

$$\begin{bmatrix} x \\ y \end{bmatrix} \leftarrow \begin{bmatrix} \min\{x + 2\alpha, 1\} \\ (y + \alpha)/\max\{x + 2\alpha, 1\} \end{bmatrix} \tag{2}$$

This update eventually converges to $\operatorname{diag}(A) = [1, 0.5]$, not the identity.

How do we fix this? To make sure that projected gradient descent will converge to the correct stationary point we must choose our descent direction to induce the most change under the correct norm: the operator 2-norm. Doing so leads to the following update,

**Lemma A.1** (Steepest descent by matrix 2-norm)**.** *The first order approximation of the steepest descent direction under the matrix operator 2-norm is given by the gradient with all non-zero singular values rescaled to be equal. That is, given a loss function $L : \mathbb{R}^{n \times m} \to \mathbb{R}$, and corresponding gradient at W, $G = \nabla L(W) = U\Lambda V^T$, one steepest descent direction is $\bar{D} = -U\mathcal{P}(\Lambda)V^T$, where the projection operator $\mathcal{P}$ sets all non-zero elements of $\Lambda$ to 1.*

*Proof.* **(Lemma A.1)** We seek the steepest descent direction,

$$\bar{D} = \underset{D}{\arg\min}\{L(W + D) \; : \; ||D||_2 = 1\}$$

Consider the first-order Taylor expansion of the loss,

$$L(W + \bar{D}) \approx L(W) + \text{Tr}(G\bar{D}^T),$$

where the trace is computing the vectorized dot product between the gradient and the descent direction. Thus, the first order approximation of the steepest descent direction seeks to minimize $\text{Tr}(G\bar{D}^T)$ subject to the 2-norm constraint on $D$. Without loss of generality, we will write $D = UU'SV'^TV^T$, then we wish to minimize $\text{Tr}(V'SU'^T\Lambda) = \text{Tr}(K\Lambda) = \sum_i \lambda_i K_{ii}$, where we write $K = V'SU'^T$, and $\lambda_i$ denotes the diagonal elements of $\Lambda$. We have reduced this to a simple constrained optimization problem where we wish to make $K_{ii}$ as negative as possible. This can be achieved when $K_{ii} = -1$ for every $\lambda_i \neq 0$. Thus, we have $\bar{D} = -U\mathcal{P}(\Lambda)V^T$. $\qquad\square$

In our experiments, we did not use the projection step with the correct steepest descent direction but instead opted to rescale the matrices by their spectral norm during the forward pass and backpropagate through this step.

## B    Examples of Lipschitz Functions

**Affine Transformations**    All affine transformations are Lipschitz functions, and their Lipschitz constant under $L_2$ is given by their spectral norm, which is the largest singular value of that linear transformation. A noteworthy special subset of linear transformations is the subset of orthogonal linear transformations. A linear transformation is considered orthogonal if it has maximal rank and all non-zero singular values equal to 1. They have the special property that an orthogonal transformation $O : \mathbb{R}^n \mapsto \mathbb{R}^m$ will preserve norm if $n \leq m$, that is, $||Ox||_2 = ||x||_2, \forall x$. In the backward pass (backpropagation), the gradient signal becomes $O^T g$, where $g$ is is the incoming gradient. Since the transpose of an orthogonal linear transformation is also orthogonal, we see that $O$ is gradient-norm-preserving for $n \geq m$.

**GroupSort**    GroupSort is a 1-Lipschitz activation function that is proposed in Anil et al. [2]. GroupSort partitions the activation vector into groups of same size and sorts the values within each group in-place. Anil et al. [2] showed that GroupSort can recover ReLU and the absolute value function. Most importantly, it addresses the capacity limitation induced by ReLU activation in Lipschitz constrained networks [12].

## C    Algorithm Complexity of Different Approaches to Enforce Lipschitz Convolution

For simplicity, we assume all the matrices to be square matrices with a size of $c \times c$ and convolution kernel to be $k \times k \times c \times c$ where $k$ is the kernel size and $c$ is the channel size. We also assume all the input has a spatial dimension of $s \times s$. In practice, the kernel size of the convolution is usually small (e.g., $k = 3$), so we consider it as a constant factor. Thus, we are mainly interested in considering the time complexity with respect to $s$ and $c$. In addition, we assume matrix multiplication of two $c \times c$ matrices takes $O(c^3)$.

**Orthogonalization using Björck and Power iteration**    First order Björck orthogonalization on a $c \times c$ matrix takes $O(c^3)$ per iteration. We use power iterations to rescale the matrix before the orthogonalization procedure to ensure convergence, which take $O(c^2)$ per iteration. Overall, the orthogonalization takes $O(c^3)$. In practice, we use 20 iterations of first order Björck and 10 power iterations.

**OSSN**    This method computes an approximated spectral norm for the convolution operator and scales down by that value. The approximated spectral norm is computed by power iteration for convolution [10], which involves convolving the convolution kernel on a tensor with the full input shape and convolving the transposed convolution kernel[1] on the full output shape per iteration. Overall, the time complexity is $O(c^3 s^2)$.

**RKO**    This method simply orthogonalizes an $c \times k^2 c$ matrix, so it takes $O(c^3)$.

**SVCM**   Singular value clipping and masking takes $O(c^3 s^2)$ per iteration (as analyzed in Sedghi et al. [22]).

**BCOP**   BCOP requires one orthogonalization of a $c \times c$ matrix and $2k-2$ orthogonalizations of $c \times \left\lceil \frac{c}{2} \right\rceil$ matrices for the symmetric projectors. Overall, the runtime for BCOP is $O(c^3)$.

| Standard | BCOP | RKO | OSSN |
|:---:|:---:|:---:|:---:|
| 0.041 | 0.138 | 0.120 | 0.113 |

Table 1: Time (in seconds) that each of the method takes for one training iteration with the "Large" architecture on one NVIDIA P100 GPU on CIFAR-10 dataset. A batch size of 128 is used.

## D   Architectural Detail Considerations for GNP Convolutional Networks

In our construction of Lipschitz convolutional networks, we restrict ourselves further to GNP convolutional networks. This has the benefit of preventing the gradient norm attenuation when each layer of a Lipschitz network is constrained, as well as giving training stability through dynamical isometry. To build a GNP network, we make every component of the network GNP. GNP convolutions have already been established by using BCOP to make orthogonal convolutions; however, there are still many more elements of a convolutional network to make GNP. An important realization is that due to dynamical isometry in GNP networks, it is no longer necessary to use the typical methods for adding stability in training, so these elements may be removed from the network. The following will discuss all the architectural decisions made for constructing networks with the GNP property while also leveraging the properties that GNP affords.

### D.1   Residual Connections

Residual connections make it difficult and unnatural to maintain a small Lipschitz constant for the network while being GNP. As well, a key feature of residual connections yields a stabler Jacobian for better training dynamics; however, the dynamical isometry property of the networks means that additional stability is not necessary. Therefore, we remove residual connections from our Lipschitz convolutional network designs.

A residual connection layer can be expressed as $g(x) = f(x) + x$, where $f$ is generally parameterized by some shallow or deep neural network. We can then bound the Lipschitz constant $\mathrm{Lip}(g)$ in terms of $\mathrm{Lip}(f)$,

$$||g(x_1) - g(x_2)|| = ||(f(x_1) + x_1) - (f(x_2) + x_2)|| \leq ||f(x_1) - f(x_2)|| + ||x_1 - x_2||$$

So we have $\mathrm{Lip}(g) \leq 1 + \mathrm{Lip}(f)$, which may be a loose bound in general, but a tighter bound is not easy to determine. To guarantee that $g$ is 1-Lipschitz, we could only do so by having $\mathrm{Lip}(f) = 0$, which means $f$ is a constant function, which obviously is not sufficiently expressive. Therefore, getting a class of 1-Lipschitz functions with residual connections is not straightforward. A possible workaround to this could be to constrain the Lipschitz constant of $f$ to 1, then halve the value after the residual connection, i.e. $g(x) = \frac{f(x)+x}{2}$. This indeed will bound $\mathrm{Lip}(g)$ by 1, but then a problem arises with gradient norm preservation. The Jacobian of $g$ would be

$$\nabla_x g = \frac{\nabla_x f + I}{2}$$

So for $g$ to be GNP almost everywhere, we would require $\nabla_x g = \frac{\nabla_x f + I}{2}$ to be orthogonal almost everywhere; however, there is no natural or well-known way to optimize over a class of non-linear functions $f$ such that $\frac{\nabla_x f + I}{2}$ is orthogonal almost everywhere. These reasons show why residual connections are hard to reconcile with Lipschitz-constrained and GNP networks.

### D.2   Zero-Padded Orthogonal Convolutions

Consider an orthogonal 1-D convolution kernel of size $2k+1$, represented by

$$[A_{-k} \quad \cdots \quad A_0 \quad \cdots \quad A_k]$$

Figure 1: Invertible Downsampling [13]

Then the corresponding Toeplitz matrix of the zero-padded convolution operation is

$$M = \begin{pmatrix} A_0 & A_1 & \dots & A_k & 0 & \dots & 0 & \dots & 0 \\ \vdots & \vdots & \ddots & \vdots & \vdots & \ddots & \vdots & \ddots & \vdots \\ A_{-k} & A_{-k+1} & \dots & A_0 & A_1 & \dots & 0 & \dots & 0 \\ \vdots & \vdots & \ddots & \vdots & \vdots & \ddots & \vdots & \ddots & \vdots \\ 0 & 0 & \dots & 0 & 0 & \dots & A_{-k} & \dots & A_0 \end{pmatrix}$$

Since the kernel is orthogonal, $MM^T = I$. This means that if $R_i$ is the $i^{th}$ block row of $M$, then $R_i R_j^T = \delta_{ij} I$. In particular, we can take the first block row, then $(k+1)^{th}$ block row (i.e. the one with $A_{-k}$ as the first element), and the last row. The first row yields the condition

$$\sum_{i=0}^{k} A_i A_i^T = I$$

The $(k+1)^{th}$ row yields

$$\sum_{i=-k}^{k} A_i A_i^T = I$$

And the last row yields

$$\sum_{i=-k}^{0} A_i A_i^T = I$$

Combining these conditions yields that $A_0 A_0^T = I$. This then implies that all other matrices must be 0. Therefore, all 1-D orthogonal kernels with zero-padding are only size 1 kernels, and so cyclic padding is used instead.

### D.3  Invertible Downsampling

The theory developed for the orthogonal convolution assumed stride 1. As such, we make sure all the convolutions are done with only stride of 1. However, since striding is an important feature in convolutional networks, we emulate it through an **invertible downsampling** layer [13] followed by a stride-1 convolution. Invertible downsampling rearranges pixels in a single channel into multiple channels so that a stride-1 convolution over the rearranged image is equivalent to a strided convolution over the original image. This layer is illustrated in Figure 1 with input channel size 1 and stride 2.

### D.4  Other Components

**Batch Normalization** Batch normalization is generally used to improve stability in training; however, it is neither 1-Lipschitz nor gradient norm preserving. Therefore, it is removed from the network.

**Linear Layers** We directly use Björck orthogonalization (See Section G) procedure to enforce orthogonal linear layers as done in Anil et al. [2].

**Activation Functions** The activation function we use is GroupSort as Anil et al. [2] found that GroupSort enhances the network capacity of Lipschitz networks compared against ReLU. In particular, we use GroupSort with a group size of 2, which is referred to as MaxMin [2] (or OPLU in Chernodub and Nowicki [7]). We use MaxMin activation because we found it to work the best in practice.

# E    Network Architectures

We describe the details for the network architectures we used in this paper and compare the number of parameters (See Table 2).

**Small**    The "Small" convolutional network contains two convolutional layers with kernel size of 4, stride 2, and channel sizes of 16 and 32 respectively, followed by two linear layers with 100 hidden units.

**Large**    The "Large" convolutional network contains four convolutional layers with kernel size of 3/4/3/4 and stride 1/2/1/2 with channel sizes of 32/32/64/64 respectively, followed by three linear layers with 512 hidden units.

**FC-3**    The "FC-3" networks refer to a 3-layer fully connected network with the number of hidden units of $1024$.

|  | **Small** | **Large** | **FC-3** | **DCGAN Critic** |
|---|---|---|---|---|
| **MNIST** | 166,406 | 1,974,762 | 2,913,290 | |
| **CIFAR-10** | 214,918 | 2,466,858 | 5,256,202 | |
| **Wasserstein Distance Estimation** | | | | 2,764,737 |

Table 2: Number of parameters for each architecture on different tasks.

**DCGAN Critic**    All of the Wassertein distance estimation experiments uses a variant of the fully convolutional critic architecture described by Radford et al. [21]. This architecture consists of 5 convolutional layers with kernel sizes of 4/4/4/4/4, strides of 2/2/2/2/1 and channel sizes of 64/128/256/512/1. We removed all the Batch Normalization layers and used either ReLU activation or MaxMin activation.

It is important to note that, in general, it is difficult to make the whole network gradient-norm-preserved because a linear transformation from a low dimensional vector to a high dimensional vector is guaranteed to lose gradient norm under some inputs. Since the aforementioned architectures mostly consist of layers that are decreasing in size, enforcing orthogonality is sufficient to enforce gradient norm preservation throughout most part of the networks (usually only the first layer is increasing in dimension).

# F    Training Details

**Provable Robustness for MNIST and CIFAR-10**    We used Adam optimizer and performed a search over $0.01$, $0.001$, $0.0001$ for the learning rate. We found $0.001$ to work the best for all experiments. We used exponential learning decay at the rate of $0.1$ per $60$ epochs. We trained the networks for 200 epochs with a batch size of 128. All of our experiments were run on NVIDIA P100 GPUs. No preprocessing was done on MNIST dataset (pixel values are between 0 and 1). For CIFAR-10, standard data augmentation is applied with random cropping (with a maximum padding of 4 pixels) and random horizontal flipping. The pixel values are between 0 and 1 with no scaling applied.

**Wasserstein Distance Estimation**    The STL-10 and CIFAR-10 GANs we used were trained using the gradient penalized Wasserstein GAN framework [3, 11]. The generator and discriminator network architectures were adapted from the ones used by Chen et al. [6]. The implementation as well as the choice of hyperparameters is based on [14]. A learning rate of 0.0002 was used for both the generator and discriminator. The gradient penalty applied on the discriminator was 50. The model was trained for 128 epoch, with a batch size of 64 using the Adam optimizer [16] with $\beta_1 = 0.5$ and $\beta_2 = 0.999$. The training was performed on NVIDIA P100 GPUs.

The DCGAN architecture [21] was used to independently compute the Wasserstein distance between the data and generator distributions of the aforementioned GAN. For the CIFAR-10 dataset, each example was upsampled to $64 \times 64$ using nearest neighbors interpolation. We used RMSprop

optimizer and performed a search over $0.1, 0.01, 0.001, 0.0001$ for the learning rate. We found that learning rate of $0.001$ works the best for models with ReLU activation and learning rate of $0.0001$ works the best MaxMin activation. The numbers reported in the table were corresponding to the best learning rates. In practice, we also observed that the training with OSSN can be unstable under high learning rate in our Wasserstein distance estimation experiments. We also used the same exponential learning decay and a batch size of 64. All the networks were trained for 25,600 iterations on NVIDIA P100 GPUs.

## G  Orthogonalization Procedure

Several ways have been proposed in the literature to orthogonalize an unconstrained matrix in a differentiable manner [18, 5]. In this work, we adopt Björck orthogonalization algorithm from Björck and Bowie [5].

The original Björck paper proposes the following iterative procedure to find the closest matrix under the metric of the Frobenius norm of the difference matrix:

$$\text{Björck}(A) = A \left( I + \frac{1}{2}Q + \frac{3}{8}Q^2 + \cdots + (-1)^p \binom{-\frac{1}{2}}{p} Q^p \right)$$

where $p \geq 1$ and $Q = I - A^T A$. This function can be iterated arbitrarily to get tighter estimates of orthogonal matrices. In all our experiments, we use $p = 1$ and iteratively apply this function 20 times.

## H  BCOP Implementation

BCOP consists of a series of block convolutions with symmetric projectors in each of the convolution component. In practice, one could use any unconstrained matrix $R \in \mathbb{R}^{n \times \lfloor \frac{n}{2} \rfloor}$ to parameterize a symmetric projector $P \in \mathbb{R}^{n \times n}$ with $\text{rank}(P) = \lfloor \frac{n}{2} \rfloor$ as follows:

$$\begin{aligned} \tilde{R} &= \text{Björck}(R) \\ P &= \tilde{R}\tilde{R}^T \end{aligned} \tag{3}$$

where $\text{Björck}$ standards for the Björck orthongalization algorithm that computes the closest orthogonal matrix (closeness in terms of the Frobenius norm) given an arbitrary input matrix [5] (See Appendix G for details on orthogonalization procedure).

For convergence guarantees of Björck, we also rescale the unconstrained matrix to be spectral norm bounded by 1 using power iteration. This rescaling procedure does not change the output orthogonal matrix at convergence because Björck is scale-invariant [5], i.e., $\text{Björck}(\alpha R) = \text{Björck}(R)$.

## I  Additional Ablation Experiments

In this section, we report the performance of some other alternative Lipschitz constrained convolutions and compare them against BCOP's performance (Table 3).

**BCOP-Fixed**  Same as BCOP method (as introduced in Section **??**) but the weights of the convolutions were frozen during training. Only the weights in the fully-connected layer are being optimized. This method was tested as a sanity check to ensure that BCOP isn't offloading all the training to the fully connected layer while the BCOP convolutional layers did little of the work, as was a phenomenon observed in Abadi et al. [1], Cox and Pinto [8].

**RK-L2NE**  Another alternative reshaped kernel (RK) method that bounds the spectral norm of a matrix as done in Qian and Wegman [20]. In particular,

$$||A||_2 \leq \max(||AA^T||_\infty, ||A^T A||_\infty)$$

We first compute the upper-bound of the spectral norm of the reshaped kernel as above and then scale the matrix down by the factor. Similar to RKO, we scale the convolution kernel (reshaped from the matrix that has a spectral norm of at most 1) down by a factor of $K$ with $K$ being the kernel size of the convolution to ensure the spectral norm of the convolution is not greater than 1. All of the computations above are done during the forward pass.

| Dataset | | | BCOP-Fixed | RK-L2NE | BCOP |
|---|---|---|---|---|---|
| **MNIST** ($\epsilon = 1.58$) | **Small** | Clean | $93.57 \pm 0.17$ | $95.85 \pm 0.12$ | $\mathbf{97.54} \pm 0.06$ |
| | | Robust | $7.51 \pm 1.18$ | $39.77 \pm 0.73$ | $\mathbf{45.84} \pm 0.90$ |
| | **Large** | Clean | $65.20 \pm 3.94$ | $96.76 \pm 0.11$ | $\mathbf{98.77} \pm 0.05$ |
| | | Robust | $0.00 \pm 0.00$ | $37.79 \pm 1.21$ | $\mathbf{56.66} \pm 0.23$ |
| **CIFAR-10** ($\epsilon = 36/255$) | **Small** | Clean | $50.61 \pm 0.65$ | $58.82 \pm 0.67$ | $\mathbf{64.53} \pm 0.30$ |
| | | Robust | $36.44 \pm 0.70$ | $44.65 \pm 0.61$ | $\mathbf{50.01} \pm 0.21$ |
| | **Large** | Clean | $47.14 \pm 0.38$ | $56.75 \pm 0.68$ | $\mathbf{72.41} \pm 0.22$ |
| | | Robust | $27.43 \pm 1.26$ | $43.40 \pm 0.46$ | $\mathbf{58.72} \pm 0.23$ |

Table 3: Clean and robust accuracy on MNIST and CIFAR-10 using different Lipschitz convolutions. The provable robust accuracy is evaluated at $\epsilon = 1.58$ for MNIST and at $\epsilon = 36/255$ for CIFAR-10. Each experiment is repeated 5 times.

| Dataset | | BCOP-Large | FC-3 | MMR-Universal |
|---|---|---|---|---|
| **MNIST** ($\epsilon = 0.3$) | Clean | $\mathbf{98.77} \pm 0.05$ | $98.71 \pm 0.02$ | $96.96$ |
| | Robust | $\mathbf{97.11} \pm 0.07$ | $97.06 \pm 0.02$ | $89.60$ |
| **CIFAR-10** ($\epsilon = 0.1$) | Clean | $\mathbf{72.41} \pm 0.22$ | $62.60 \pm 0.39$ | $53.04$ |
| | Robust | $\mathbf{62.97} \pm 0.30$ | $53.67 \pm 0.29$ | $46.40$ |

Table 4: Comparison of our convolutional networks against the provable robust model from Croce and Hein [9]. The numbers for MMR-Universal are directly obtained from Table 1 in their paper [9].

| Dataset | | BCOP-Large | FC-3 | QW-3 | QW-4 |
|---|---|---|---|---|---|
| **MNIST** ($\epsilon = 1.58$) | Clean | $\mathbf{98.77} \pm 0.05$ | $98.71 \pm 0.02$ | $98.65$ | $98.23$ |
| | Robust | $\mathbf{56.66} \pm 0.23$ | $54.46 \pm 0.30$ | $42.13$ | $27.59$ |
| | PGD | $86.86 \pm 0.25$ | $81.96 \pm 0.16$ | $\mathbf{86.86}$ | $86.25$ |
| | FGSM | $\mathbf{86.93} \pm 0.20$ | $83.64 \pm 0.10$ | $85.83$ | $84.17$ |
| **CIFAR-10** ($\epsilon = 36/255$) | Clean | $72.41 \pm 0.22$ | $62.60 \pm 0.39$ | $\mathbf{79.15}$ | $77.15$ |
| | Robust | $\mathbf{58.72} \pm 0.23$ | $49.97 \pm 0.35$ | $44.46$ | $31.41$ |
| | PGD | $64.39 \pm 0.26$ | $50.05 \pm 0.36$ | $\mathbf{72.07}$ | $71.89$ |
| | FGSM | $64.53 \pm 0.25$ | $50.21 \pm 0.34$ | $\mathbf{72.11}$ | $71.92$ |

Table 5: Comparison of our convolutional networks against the provable robust models (QW-3 for Model-3 and QW-4 for Model-4) from Qian and Wegman [20]. The model weights are directly downloaded from their official website.

## J   Comparison to Other Baselines for Provable Adversarial Robustness on MNIST and CIFAR-10

We also compare the performance of the GNP networks with another baseline (MMR-Universal) from Croce and Hein [9]. The provable robustness results are summarized in Table 4. Both our "Small" and "Large" model outperform the "Small" model from Croce and Hein [9] in terms of clean accuracy and robust accuracy. The fully connected Lipschitz constrained network baseline from Anil et al. [2] achieves slightly better performance than convolutional networks we use for MNIST, but does much worse for CIFAR-10.

In addition, we also compare against the models from Qian and Wegman [20]. To encourage robustness against adversary, they proposed to use (norm) non-expansive operation only, which equivalently enforces the Lipschitz constant of the network to be at most 1. Instead of using a single network to predict the logits for all the classes, Qian and Wegman [20] uses a separate 1-Lipschitz network to predict the logit for each of the 10 classes in both MNIST and CIFAR-10 datasets. It can

be shown that we can at best certify **x** is robustly classified if

$$\mathcal{M}_f(\mathbf{x}) > 2\epsilon. \tag{4}$$

We report the provably robust accuracy (using the certification criterion presented in Equation 4) and the robust accuracy against two gradient-based attacks in Table 5. It is important to note that the QW models are designed to be robust under empirical attacks instead of obtaining certification of robustness. Also, the perturbation sizes used in their original paper are much larger than the ones that we are focusing on.

## K   Topology of of 1-D Orthogonal Convolution Kernel

In this section, we introduce the following theorem:

**Theorem 1** (Connected Components of 1-D Orthogonal Convolution). *The 1-D orthogonal convolution space is compact and has $2(K-1)n+2$ connected components, where $K$ is the kernel size and $n$ is the number of channels.*

To prove this theorem, we first discuss important properties of symmetric projectors which is of central importance of the proof in Appendix K.1. Followed by the discussion, we focus on finding the connected components of a subset of orthogonal convolution kernels – special orthogonal convolution kernels (SOCK) – in Appendix K.2. Finally, we show how the connected components of SOCK can be trivially extended to the connected components of orthogonal convolution kernel.

### K.1   Background: Symmetric Projector

Before we discuss the topology of orthogonal convolution kernels, we first review some basic properties of symmetric projectors.

**Definition 1.** *$\mathbb{P}(n)$ is the space of all $n \times n$ symmetric projectors. Formally,*

$$\mathbb{P}(n) = \{P | P^2 = P^T = P, P \in \mathbb{R}^{n \times n}\}$$

*We also denote the space of rank-$k$ symmetric projectors as $\mathbb{P}(n,k)$:*

$$\mathbb{P}(n,k) = \{P | \operatorname{rank}(P) = k, P \in \mathbb{P}(n)\}$$

**Remark 1.1.** *Followed from the definition of symmetric projectors, we can make a few observations:*

1. *Each symmetric projector can be identified with an orthogonal projection onto a linear subspace.*

2. *The range operator of matrix is a bijection map between $\mathbb{P}(n)$ and all linear subspaces of $\mathbb{R}^n$. In particular, the map bijectively sends $\mathbb{P}(n,k)$ to $\operatorname{Gr}(k, \mathbb{R}^n)$ (or Grassmannian manifold), which is defined as the set of all $k$-dimensional subspaces of $\mathbb{R}^n$ [19].*

**Theorem 4** (Symmetric Projectors and Grassmannian Manifold). *$\mathbb{P}(n,k)$ and $\operatorname{Gr}(k, \mathbb{R}^n)$ are homeomorphic.*

**Remark 4.1.** *The homeomorphism allows us to inherit properties from Grassmannian manifold:*

1. *$\mathbb{P}(n,k)$ is compact and path connected*

2. *$\mathbb{P}(n,k)$ is disjoint from $\mathbb{P}(n,k')$ for $k \neq k'$*

3. *By compactness and disjointness above, $\mathbb{P}(n,k) \cup \mathbb{P}(n,k')$ for $k \neq k'$ is path disconnected.*

4. *The dimensionality of $\mathbb{P}(n,k)$ is $k(n-k)$, which is maximized when $k = \lceil \frac{n}{2} \rceil, \lfloor \frac{n}{2} \rfloor$.*

### K.2   Connected Components of 1-D Special Orthogonal Convolution Kernels (SOCK)

We will be using the results discussed in the previous section to identify the connected components of orthogonal convolution kernel (OCK) submanifold. In the rest of this section, we put our focus on

the case of special orthogonal convolution kernel (SOCK) where the convolution operator belongs to the special orthogonal group. Because of the symmetry between the orthogonal transformation with $\det = 1$ and $\det = -1$, we can trivially determine all the connected components of OCK from the connected components of SOCK as there is no intersection between the components of OCK with $\det(H) = -1$ and $\det(H) = 1$. For the rest of this section, we put our focus on a representation of SOCK:

**Definition 2.** *A **1-D special orthogonal convolution kernel (SOCK) submanifold**, denoted by $\mathcal{C}(r_1, r_2, \cdots, r_{K-1})$, is a submanifold of $\mathbb{R}^{n \times nK}$ that can be represented by*

$$A = H \square [P_1 \quad I - P_1] \square \cdots \square [P_{K-1} \quad I - P_{K-1}] \tag{5}$$

*where*

- *"$\square$" is the block convolution operator that convolves one block matrix with another:*

$$[X_1 \quad X_2 \quad \cdots \quad X_p] \square [Y_1 \quad Y_2 \quad \cdots \quad Y_q] = [Z_1 \quad Z_2 \quad \cdots \quad Z_{p+q-1}]$$

  *with $Z_i = \sum_j X_j Y_{i-j}$, where the out-of-range elements being all zero (e.g., $X_{<1} = 0, X_{>p} = 0, Y_{<1} = 0, Y_{>q} = 0$).*

- *$P_i \in \mathbb{P}(n, r_i), \forall i$. We refer $r = (r_1, r_2, \cdots, r_{K-1})$ as the rank tuple of the SOCK submanifold $\mathcal{C}(r_1, r_2, \cdots, r_{K-1})$, or $\mathcal{C}(r)$ in short.*

- *$H \in SO(n)$.*

*We shorthand the representation described above as $A = \mathcal{A}(P_1, \cdots, P_{K-1}, H)$. Formally,*

$$\mathcal{C}(r) = \mathcal{C}(r_1, r_2, \cdots, r_{K-1}) = \{A | A = \mathcal{A}(P_1, \cdots, P_{K-1}, H), P_i \in \mathbb{P}(n, r_i), H \in SO(n)\}.$$

*We can also define*

$$\mathcal{C} = \{A | A = \mathcal{A}(P_1, \cdots, P_{K-1}, H), P_i \in \mathbb{P}(n), H \in SO(n)\}.$$

**Theorem 5** (Completeness (Theorem 2 of Kautsky and Turcajová [15]))**.** *$\mathcal{C}$ is the space of 1-D special orthogonal convolution kernels.*

**Definition 3.** *The **canonical rank tuple** of special orthogonal convolution is defined as*

$$r^{(k)} = \left( r_1^{(k)}, r_2^{(k)}, \cdots, r_{K-1}^{(k)} \right)$$

*with the following conditions:*

1. *$\sum_i r_i^{(k)} = k$*

2. *$\left| r_i^{(k)} - r_{i'}^{(k)} \right| \leq 1, \forall i, i'$*

3. *$r_i^{(k)} \leq r_{i+1}^{(k)}, \forall i$*

*Intuitively, these conditions enforce the ranks to be most balanced while having their sum equal to $k$. The last condition makes $r^{(k)}$ unique for each $k$. Since each rank tuple defines a SOCK submanifold, we can define the **canonical SOCK submanifold** as follows:*

$$\mathcal{C}_k = \mathcal{C}\left( r^{(k)} \right) = \mathcal{C}\left( r_1^{(k)}, r_2^{(k)}, \cdots, r_{K-1}^{(k)} \right).$$

**Theorem 6** (Space of 1-D Convolution Kernel)**.** *1-D special orthogonal convolution space is compact and it consists of $(K-1)n + 1$ distinct canonical SOCK submanifolds as its connected components: $\mathcal{C}_0, \mathcal{C}_1, \cdots, \mathcal{C}_{(K-1)n}$.*

The main idea of proving this theorem is to show that **any SOCK with the sum of its rank tuple equal to $k$ is a subset of $\mathcal{C}_k$.** Our proof of the theorem is divided into the following three steps:

1. **Equivalent SOCK Construction**: We identify the changes that we can make to the rank of the symmetric projectors in the representation (Equation 5) without changing the kernel that they represent, and find pairs of SOCK submanifolds in which one fully contains another (Appendix K.2.1).

2. **Dominance of Canonical SOCK Submanifold**: We prove that the canonical SOCK sub-manifolds fully contain all other SOCK submanifolds using the relationship between SOCK submanifolds identified above. A consequence of this result is that the union of the canonical SOCK submanifolds $\mathcal{C}_k$ is complete (Appendix K.2.2).

3. **Connected Components of $\mathcal{C}$ are Canonical SOCK submanifolds**: Given the result in Step 2, we complete the proof of Theorem 6 by showing that the canonical SOCK submanifolds $\mathcal{C}_k$ are compact and disjoint, and hence the number of connected components of $\mathcal{C}$ is $(K-1)n+1$, which is the number of distinct canonical SOCK submanifolds. (Appendix K.2.3).

### K.2.1 Equivalent SOCK Construction

Now, we introduce one important property of symmetric projectors that guide the construction of equivalent representations with changes in the symmetric projectors.

**Lemma 6.1** (Symmetric Projector Pair Equivalence under Product and Sum). *For all $P_1' \in \mathbb{P}(n, k_1 - 1)$ and $P_2' \in \mathbb{P}(n, k_2+1)$ with $1 \le k_1 \le k_2+1$, there always exists $P_1 \in \mathbb{P}(n, k_1)$ and $P_2 \in \mathbb{P}(n, k_2)$ such that $P_1' + P_2' = P_1 + P_2$, $P_1'P_2' = P_1P_2$, $P_2'P_1' = P_2P_1$.*

*Proof.* **(Lemma 6.1)** Let $Q_1'$ be the range of $P_1'$ and $Q_2'$ be the range of $P_2'$. We observe that $\dim(Q_1' + Q_2'^\perp) \le \dim(Q_1') + \dim(Q_2'^\perp) = k_1 - 1 + n - k_2 - 1 \le n - 1$. Therefore, there always exists a unit vector $\mathbf{x}$ in $Q_2' \cap Q_1'^\perp$. Then, we can find a orthonormal basis decomposition of $Q_2'$ that contains $\mathbf{x}$: $Q_2' = \mathrm{span}(\{\mathbf{x}_1, \mathbf{x}_2, \cdots, \mathbf{x}_{k_2}, \mathbf{x}\})$. Then, we can construct the linear subspaces $Q_1$ and $Q_2$ as follows:

$$Q_1 = Q_1' + \mathrm{span}(\{\mathbf{x}\}) \tag{6}$$
$$Q_2 = \mathrm{span}(\{\mathbf{x}_1, \mathbf{x}_2, \cdots, \mathbf{x}_{k_2}\}) \tag{7}$$

Now, we can define $P_1$ and $P_2$ to be the symmetric projectors whose range are $Q_1$ and $Q_2$ respectively. By the construction of $\mathbf{x}$, it is clear that $Q_1$ has one more dimension than $Q_1'$ and $Q_2$ has one less dimension than $Q_2'$, which makes $\mathrm{rank}(P_1) = k_1$ and $\mathrm{rank}(P_2) = k_2$. We are then only left to prove that $P_1 + P_2 = P_1' + P_2'$, $P_1'P_2' = P_1P_2$, and $P_2'P_1' = P_2P_1$.

We first observe that the orthogonal projection $P_1$ can be decomposed into the sum of two orthogonal projections onto two orthogonal subspaces: $P_1 = P_1' + \mathbf{x}\mathbf{x}^T$ where the first projection is onto $Q_1'$ and the second projection is onto $\mathrm{span}(\{\mathbf{x}\})$ with $Q_1' \perp \mathrm{span}(\{\mathbf{x}\})$. Similarly, $Q_2$ and $\mathrm{span}(\{\mathbf{x}\})$ are orthogonal subspaces and $Q_2' = Q_2 + \mathbf{x}$ from Equation 7. Decomposing $P_2'$ leads to $P_2 = P_2' - \mathbf{x}\mathbf{x}^T$.

From here it is clear that

$$P_1 + P_2 = P_1' + \mathbf{x}\mathbf{x}^T + P_2' - \mathbf{x}\mathbf{x}^T = P_1' + P_2'$$

Since $\mathbf{x} \in Q_1'^\perp$ and $\mathbf{x} \in Q_2'$, we have $P_1'\mathbf{x} = \mathbf{0}$, $P_2'\mathbf{x} = \mathbf{x}$. This allows us to complete the the proof:

$$P_1 P_2 = (P_1' + \mathbf{x}\mathbf{x}^T)(P_2' - \mathbf{x}\mathbf{x}^T) = P_1'P_2' + \mathbf{x}(P_2'\mathbf{x})^T - \mathbf{x}\mathbf{x}^T = P_1'P_2',$$
$$P_2 P_1 = (P_2' - \mathbf{x}\mathbf{x}^T)(P_1' + \mathbf{x}\mathbf{x}^T) = P_2'P_1' + \mathbf{x}\mathbf{x}^T - \mathbf{x}(P_1'\mathbf{x})^T - \mathbf{x}\mathbf{x}^T = P_2'P_1'$$

$\square$

**Lemma 6.2** (Equivalent SOCK Construction). $\mathcal{A}(P_1, \cdots, P_i, P_{i+1}, \cdots, P_{K-1}, H) = \mathcal{A}(P_1, \cdots, P_i', P_{i+1}', \cdots, P_{K-1}, H')$ *iff* $P_i + P_{i+1} = P_i' + P_{i+1}'$, $P_iP_{i+1} = P_i'P_{i+1}'$, *and* $H = H'$.

The proof of Lemma 6.2 is in Appendix L.

Now, we can find pairs of SOCK submanifold where one fully contains another.

**Lemma 6.3** (Balanced Rank Dominates). $\mathcal{C}(r_1, r_2, \cdots, r_i - 1, r_{i+1} + 1, \cdots r_{K-1}) \subseteq \mathcal{C}(r_1, r_2, \cdots r_{K-1})$ *when* $r_i \le r_{i+1} + 1$ *and* $\mathcal{C}(r_1, r_2, \cdots, r_i + 1, r_{i+1} - 1, \cdots r_{K-1}) \subseteq \mathcal{C}(r_1, r_2, \cdots r_{K-1})$ *when* $r_{i+1} \le r_i + 1$.

*Proof.* In the case where $r_i \leq r_{i+1} + 1$, suppose $P_i \in \mathbb{P}(n, r_i - 1)$ and $P_{i+1} \in \mathbb{P}(n, r_{i+1} + 1)$. By Lemma 6.1, there exists $P_i' \in \mathbb{P}(n, r_i)$ and $P_{i+1}' \in \mathbb{P}(n, r_{i+1})$ such that $P_i + P_{i+1} = P_i' + P_{i+1}'$, $P_i P_{i+1} = P_i' P_{i+1}'$. By Lemma 6.2, $\mathcal{A}(P_1, \cdots, P_{K-1}, H) = \mathcal{A}(P_1, \cdots, P_i', P_{i+1}', \cdots P_{K-1}, H)$. Since this holds regardless of the choice of $P$'s, we can conclude $\mathcal{C}(r_1, r_2, \cdots, r_i - 1, r_{i+1} + 1, \cdots r_{K-1}) \subseteq \mathcal{C}(r_1, r_2, \cdots r_{K-1})$.

In the case where $r_{i+1} \leq r_i + 1$, the same proof holds by symmetry. $\qquad\square$

**Lemma 6.4** (Rank Balancing). $\mathcal{C}(r_1, r_2, \cdots, r_i - \delta, \cdots, r_j + \delta, \cdots, r_{K-1}) \subseteq \mathcal{C}(r_1, r_2, \cdots, r_i, \cdots, r_j, \cdots, r_{K-1})$ *if* $r_i = r_p = r_j$ *for* $i < p < j$ *and* $\delta \in \{-1, 1\}$.

*Proof.* The Lemma can be proven by induction on $i$ facilitated by Lemma 6.3. $\qquad\square$

### K.2.2  Dominance of Canonical SOCK Submanifold

Using all the subspace relations that we identify above (Lemma 6.3 and 6.4) repeatedly from any SOCK submanifold, we can find another SOCK submanifold with a more balanced rank tuple that fully contains the former submanifold. To formalize the notion of "balanced rank tuple", we introduce **imbalance score function**:

**Definition 4.** *The imbalance score for an special orthogonal convolution subspace* $\mathcal{C}(r)$ *is*

$$f(r) = \sum_i (r_i)^2 \tag{8}$$

*For any rank tuple that has the minimum imbalance score under the constraint that the sum of them is* $k$, *we call it a* ***balanced rank tuple*** *with sum* $k$. *It is clear that the condition of having the minimum imbalance score is also equivalent to having the following condition:*

$$|r_i - r_j| \leq 1, \forall i, j$$

**Lemma 6.5** (SOCKs with Balanced Rank Tuple are Equiavlent). *Let* $r$ *be a balanced rank tuple. Then,* $\mathcal{C}(r) = \mathcal{C}_k$ *with* $k = \sum_i r_i$.

*Proof.* Any balanced rank tuple has a rank difference of at most 1. Then, we can use Lemma 6.3 to swap any two adjacent ranks such that

$$\mathcal{C}(r_1, \cdots, c_{i+1}, c_i, \cdots, r_{K-1}) \subseteq \mathcal{C}(r_1, \cdots, c_i, c_{i+1}, \cdots, r_{K-1}).$$

Now, if we apply the swap again, we will get

$$\mathcal{C}(r_1, \cdots, c_{i+1}, c_i, \cdots, r_{K-1}) \supseteq \mathcal{C}(r_1, \cdots, c_i, c_{i+1}, \cdots, r_{K-1}),$$

which means

$$\mathcal{C}(r_1, \cdots, c_{i+1}, c_i, \cdots, r_{K-1}) = \mathcal{C}(r_1, \cdots, c_i, c_{i+1}, \cdots, r_{K-1}).$$

From here, it is obvious that we can propagate the equivalance relationship from any balanced rank tuple to a canonical rank tuple by performing bubble sort while the sum of the ranks remains the same. Therefore, any SOCK submanifold with its sum of ranks equals to $k$ would be equivalent to $\mathcal{C}_k$. $\qquad\square$

Since a SOCK with balanced rank tuple is equiavlent to its corresponding canonical SOCK, we are only left to show that the imbalance score function can always be decreased until the rank is most balanced.

**Lemma 6.6** (Balancing Subspace Dominates). *Given a rank tuple* $r$, *if there exists* $i, j$ *such that* $|r_i - r_j| \geq 2$, *then there exists* $\mathcal{C}(r') \supseteq \mathcal{C}(r)$ *with* $\sum_i r_i' = \sum_i r_i$ *such that* $f(r) > f(r')$.

*Proof.* (**Lemma 6.6**) Let $i < j$ be the closest pair of points such that $|r_i - r_j| \geq 2$. Without loss of generality, we assume $r_i < r_j$ since the same proof below would hold for the case with $r_i > r_j$ by symmetry. Now, consider the following rank tuple

$$r' = (r_1, \cdots, r_i + 1, \cdots, r_j - 1, \cdots, r_{K-1}).$$

It is clear that

$$f(r') - f(r) = 2(r_i - r_j) < 0.$$

Now, we are left to show that $\mathcal{C}(r') \supseteq \mathcal{C}(r)$.

**Case 1** Assume the two ranks are adjacent, or $i + 1 = j$. By Lemma 6.3, $\mathcal{C}(r) \subseteq \mathcal{C}(r')$.

**Case 2** Assume the two ranks are not adjacent, or $i + 1 < j$. We must have $r_p = r_i + 1 = r_j - 1$ for $i < p < j$. Otherwise, the $i$ and $j$ would no longer be the closest pair such that $|r_i - r_j| \geq 2$. We can then apply Lemma 6.4 to get $\mathcal{C}(r) \subseteq \mathcal{C}(r')$. $\qquad\square$

**Lemma 6.7** (Canonical Submanifold Dominates). $\mathcal{C}(r') \subseteq \mathcal{C}_k$ if $\sum_p r'_p = k$.

*Proof.* **(Lemma 6.7)** We prove this Lemma by dividing it up into the following two cases:

**Case 1** Assume there is no $i, j$ such that $|r'_i - r'_j| \geq 2$, the rank tuple $r'$ is balanced by Definition 4. Then, by Lemma 6.5, $\mathcal{C}(r') = \mathcal{C}_k$.

**Case 2** Assume there exists $i, j$ such that $|r'_i - r'_j| \geq 2$. Then by Lemma 6.6, we can always find a special orthogonal convolution subspace that contains the current subspace $\mathcal{C}(r')$ without changing the sum of the ranks and with the strictly decreased imbalance score. Since the imbalance score takes on natural numbers, iteratively applying the Lemma to decrease the imbalance score must eventually terminate. When it does, it will yield the subspace $\mathcal{C}(r^*)$ which contains the $\mathcal{C}(r')$ and has $|r_i^* - r_j^*| < 2$ for all $i, j$. We are left to show that $\mathcal{C}(r^*) \subseteq \mathcal{C}_k$, which is shown in the first case. $\qquad\square$

**Corollary 6.7.1** (The union of all canonical subspaces is complete ). $\mathcal{C} = \mathcal{C}_0 \cup \mathcal{C}_1 \cup \cdots \cup \mathcal{C}_{(K-1)n}$.

*Proof.* **(Corollary 6.7.1)** $\mathcal{C}$ is the union of all SOCK submanifolds, where each of them belong to at least one of $\mathcal{C}_k$ by Lemma 6.7. From here, it is clear that the Corollary holds. $\qquad\square$

### K.2.3 Connected Components of $\mathcal{C}$ are Canonical SOCK Submanifolds

To fill in the final piece of proving Theorem 6, we need to prove that $\mathcal{C}_k$ are disjoint and compact. We first prove the disjointness by expressing the sum of symmetric projectors used to construct the kernel as a linear combination of kernel elements $\{A_1, A_2, \cdots, A_K\}$ below. The proof of Lemma 6.8 is in Appendix L.

**Lemma 6.8** (Kernel Element Decomposition). *If* $A = \mathcal{A}(P_1, \cdots, P_{K-1}, H)$, *then* $A_j = \sum_{i=0}^{j} a_{K,j,i} B_i$, *where* $a_{K,j,i} = (-1)^{j-i} \binom{(K-1)-i}{j-i}$ *for* $i \leq j \leq K - 1$ *and 0 otherwise, and* $B_k = \sum_{\delta'_1, \delta'_2, \cdots, \delta'_{K-1} | \sum_i \delta'_i = k} H \prod_{1 \leq i \leq K-1} [(1 - \delta'_i) P_i + \delta'_i I]$

**Corollary 6.8.1** (Triangular Map between $A$ and $B$). $B_j = A_j - \sum_{k=0}^{j-1} (-1)^{j-k} \binom{(K-1)-k}{j-k} B_k$.

*Proof.* **(Corollary 6.8.1)** The expression can be obtained by simply rearrange the expression of $A_j$ from Lemma 6.8. $\qquad\square$

*Proof.* **(Theorem 6)** From Corollary 6.8.1, we can recursively expand out the terms on the right to express $B_j$ as a linear combination of $\{H^T A_1, H^T A_2, \cdots, H^T A_{K-1}\}$. Formally, $B_j = \sum_k (w_{jk} H^T A_k)$ for some $w_{ji}$. We can then define a continuous function $g$, $g(A) = \text{Tr} \sum_k (w_i H^T A_i) = \text{Tr } H^T B_{K-2} = \sum_i \text{Tr}(H^T H P_i) = \sum_i \text{Tr } P_i = \sum_i r_i = r$. Therefore $g(\mathcal{C}_r) = \{r\}$ and $g(\mathcal{C}_{r'}) = \{r'\}$, so $\mathcal{C}_r, \mathcal{C}_{r'}$ must be disjoint if $r \neq r'$.

Next, we prove compactness of each $\mathcal{C}(r)$. We first observe that symmetric projector submanifold, $\mathbb{P}(n, k)$, is compact and path connected for any $k$ (Remark 4.1). $\mathcal{C}(r)$ is the image of $\mathcal{A}$ under these $K - 1$ sets of $P_i$'s and the set of special orthogonal matrices $H$'s. All of these sets are compact and path-connected and $\mathcal{A}$ is continuous; therefore $\mathcal{C}(r)$ is path connected and compact.

Finally, since $\mathcal{C}_r$ and $\mathcal{C}_{r'}$ are compact and disjoint, $\mathcal{C}_r \cup \mathcal{C}_{r'}$ is path-disconnected for $r \neq r'$. Combining this with the connectedness of each individual $\mathcal{C}_r$ as well as completeness of the $\{\mathcal{C}_r\}$ (Corollary 6.7.1), we can conclude that the $\mathcal{C}_r$'s are the connected components of $\mathcal{C}$, so there are $(K - 1)n + 1$ total disconnected components. $\qquad\square$

**Theorem 1** (Connected Components of 1-D Orthogonal Convolution). *The 1-D orthogonal convolution space is compact and has $2(K - 1)n + 2$ connected components, where $K$ is the kernel size and $n$ is the number of channels.*

*Proof.* From Theorem 6, $C$ is the union of all SOCK submanifolds which contain $(K-1)n+1$ connected components. The other subset of orthogonal convolution kernels that we omitted when considering SOCK can be simply obtained by negating one row of the orthogonal matrix in the SOCK representation (Equation 5). Because there exists no continuous path from the components with determinants of $-1$ to the components with determinants of 1. The number of connected components in orthogonal convolution kernels is therefore doubled, which is $2(K-1)n+2$. □

## L  Additional Proofs

**Lemma 6.2** (Equivalent SOCK Construction). $\mathcal{A}(P_1, \cdots, P_i, P_{i+1}, \cdots, P_{K-1}, H) = \mathcal{A}(P_1, \cdots, P_i', P_{i+1}', \cdots, P_{K-1}, H')$ *iff* $P_i + P_{i+1} = P_i' + P_{i+1}'$, $P_i P_{i+1} = P_i' P_{i+1}'$, *and* $H = H'$.

*Proof.* **(Lemma 6.2)** Let

- $A = \mathcal{A}(P_1, \cdots, P_i, P_{i+1}, \cdots, P_{K-1}, H)$

- $A' = \mathcal{A}(P_1, \cdots, P_i', P_{i+1}', \cdots, P_{K-1}, H')$

- $Q$ be a function of a set of binary variables $(\delta_j)$ that control which factor ($P_j$ or $I - P_j$) appears in the product of matrices in the function output:

$$Q(\delta_1, \delta_2, \cdots, \delta_{K-1}) = H \prod_{1 \le j \le K-1} [(1 - \delta_j)P_j + \delta_j(1 - P_j)],$$

  where $\delta_j \in \{0, 1\}$.

- $Q'$ be the function in similar form as $Q$, but the summation elements are constructed with respect to the new kernel $A'$.

Using the $Q$ function above, we can represent $A$ in an alternate form

$$A_j = \sum_{\delta_1, \delta_2, \cdots \delta_{K-1} | \sum_k \delta_k = j-1} Q(\delta_1, \delta_2, \cdots, \delta_{K-1})$$

where $1 \le j \le K - 1$. This form will make our proof below more convenient.

The proof is divided into the following two steps (forward direction and backward direction):

**"⇒" Direction**

$$A = A' \Rightarrow P_i + P_{i+1} = P_i' + P_{i+1}', P_i P_{i+1} = P_i' P_{i+1}', H = H'$$

We start by summing all the elements of $A$ and $A'$ to show $H = H'$:

$$\sum_j A_j = \sum_j \sum_{\delta_1, \delta_2, \cdots, \delta_{K-1} | \sum_k \delta_k = j-1} Q(\delta_1, \delta_2, \cdots, \delta_{K-1})$$

$$= \sum_{\delta_1, \delta_2, \cdots, \delta_{K-1}} Q(\delta_1, \delta_2, \cdots, \delta_{K-1})$$

$$= \sum_{\delta_1, \delta_2, \cdots, \delta_{K-1}} H \prod_{1 \le j \le K-1} [(1 - \delta_j)P_j + \delta_j(I - P_j)]$$

$$= H \prod_{1 \le j \le K-1} [P_j + (I - P_j)]$$

$$= H$$

The same process for $A'$ will show $\sum_j A_j' = H'$. Thus, $H = H'$. We are still left to show $P_i + P_{i+1} = P_i' + P_{i+1}'$ and $P_i P_{i+1} = P_i' P_{i+1}'$. Now, we will consider $A$ and $A'$ in their alternate form,

$$A = H \square [P_1 \quad I - P_1] \square \cdots \square [P_{K-1} \quad I - P_{K-1}]$$
$$A' = H' \square [P_1 \quad I - P_1] \square \cdots [P_i' \quad I - P_i'] \square [P_{i+1}' \quad I - P_{i+1}'] \square \cdots \square [P_{K-1} \quad I - P_{K-1}]$$

First, we do a left block convolution by $H^T$ to obtain

$$H^T \square A = H^T \square H \square [P_1 \quad I - P_1] \square \cdots \square [P_{K-1} \quad I - P_{K-1}]$$
$$= [P_1 \quad I - P_1] \square \cdots \square [P_{K-1} \quad I - P_{K-1}]$$
$$H^T \square A' = H^T \square H' \square [P_1 \quad I - P_1] \square \cdots \square [P_i' \quad I - P_i'] \square [P_{i+1}' \quad I - P_{i+1}'] \square \cdots \square [P_{K-1} \quad I - P_{K-1}]$$
$$= [P_1 \quad I - P_1] \square \cdots \square [P_i' \quad I - P_i'] \square [P_{i+1}' \quad I - P_{i+1}'] \square \cdots \square [P_{K-1} \quad I - P_{K-1}]$$

Now, since only the $i^{th}$ and $(i + 1)^{th}$ convolutions differ between these sequences, we can iteratively convolve away every other convolution by left/right-convolving with the inverse of the left/right-most element. On the left, we would begin by convolving with $[I - P_1 \quad P_1]$ and on the right by $[I - P_{K-1} \quad P_{K-1}]$. We can then continue performing the left/right convolution to repeatedly cancel out the left/right-most element until we are left with the two terms with index of $i$ and $i + 1$ as follows:

$$[P_i \quad I - P_i] \square [P_{i+1} \quad I - P_{i+1}] = [P_i' \quad I - P_i'] \square [P_{i+1}' \quad I - P_{i+1}']$$

Expanding out the convolution and re-arranging the equation gives $P_i + P_{i+1} = P_i' + P_{i+1}'$ and $P_i P_{i+1} = P_i' P_{i+1}'$.

**"$\Leftarrow$" Direction**

$$P_i + P_{i+1} = P_i' + P_{i+1}', P_i P_{i+1} = P_i' P_{i+1}', H = H' \Rightarrow A = A'$$

To prove the backward direction, we can simply invert the proof in the forward direction:

$P_i + P_{i+1} = P_i' + P_{i+1}'$ and $P_i P_{i+1} = P_i' P_{i+1}'$ implies that

$$[P_i \quad I - P_i] \square [P_{i+1} \quad I - P_{i+1}] = [P_i' \quad I - P_i'] \square [P_{i+1}' \quad I - P_{i+1}']$$

Then, it is clear that

$$H \square [P_1 \quad I - P_1] \square \cdots \square [P_i \quad I - P_i] \square [P_{i+1} \quad I - P_{i+1}] \square \cdots \square [P_{K-1} \quad I - P_{K-1}]$$
$$= H' \square [P_1 \quad I - P_1] \square \cdots \square [P_i' \quad I - P_i'] \square [P_{i+1}' \quad I - P_{i+1}'] \square \cdots \square [P_{K-1} \quad I - P_{K-1}],$$

which yields $A = A'$ as what we needed. $\qquad \square$

**Lemma 6.8** (Kernel Element Decomposition). *If* $A = \mathcal{A}(P_1, \cdots, P_{K-1}, H)$, *then* $A_j = \sum_{i=0}^{j} a_{K,j,i} B_i$, *where* $a_{K,j,i} = (-1)^{j-i} \binom{(K-1)-i}{j-i}$ *for* $i \leq j \leq K - 1$ *and 0 otherwise, and* $B_k = \sum_{\delta_1', \delta_2', \cdots, \delta_{K-1}' | \sum_i \delta_i' = k} H \prod_{1 \leq i \leq K-1} [(1 - \delta_i') P_i + \delta_i' I]$

*Proof.* **(Lemma 6.8)**

We will show this result by considering the following form of $A_j$,

$$A_j = \sum_{\delta_1, \delta_2, \cdots, \delta_{K-1} | \sum_i \delta_i = j} Q(\delta_1, \delta_2, \cdots, \delta_{K-1})$$

where $Q$ is given by

$$Q(\delta_1, \delta_2, \cdots, \delta_{K-1}) = H \prod_{1 \leq i \leq K-1} [(1 - \delta_i) P_i + \delta_i (I - P_i)]$$

and $\delta_i \in \{0, 1\}$.

Every summand of $A_j$ can be expanded into the form

$$Q(\delta_1, \delta_2, \cdots, \delta_{K-1}) = \sum_{\delta_1', \delta_2', \cdots, \delta_{K-1}' | \sum_i \delta_i' \leq j} a_{\delta_1', \cdots, \delta_{K-1}'} H \prod_{1 \leq i \leq K-1} [(1 - \delta_i') P_i + \delta_i' I]$$

for some coefficients $\{a_{\delta_1', \cdots, \delta_{K-1}'}\}$, and we will provide a closed form for these coefficients.

When $\delta_i = 0$, the $i^{th}$ factor of $Q(\delta_1, \delta_2, \cdots, \delta_{K-1})$ is just $P_i$, and this term does not expand. This means that all summands in the expansion must contain $P_i$. Therefore, $a_{\delta_1', \cdots, \delta_{K-1}'} = 0$ whenever

$\delta_i' = 1$. When $\delta_i = 1$, the $i^{th}$ factor is instead $I - P_i$, which would generate two parts in the expansion, one with $I$, and the other with $-P_i$. Therefore, if $\delta_i = 1$, for a given $a_{\delta_1', \cdots, \delta_{K-1}'}$, the $i^{th}$ factor provides a factor of -1 to the coefficient if $\delta_i' = 0$, and provides a factor of 1 otherwise. This means that the final coefficient will be $(-1)^n$, where $n$ is the number of positions $i$ where $\delta_i = 1$ but $\delta_i' = 0$. Thus the closed form is

$$a_{\delta_1', \cdots, \delta_{K-1}'} = \begin{cases} 0, & \text{if } \exists\, i \text{ such that } \delta_i = 0, \delta_i' = 1 \\ (-1)^{\sum_i \delta_i - \delta_i'}, & \text{otherwise} \end{cases}$$

Notice that $\sum_i \delta_i$ is constant for all summands of $A_j$, so $a_{\delta_1', \cdots, \delta_{K-1}'}$ is constant over all summands in which it is non-zero. Therefore, we can find how many summands in which this coefficient is non-zero, and that will give us the total coefficient for $H \prod_{1 \leq i \leq K-1} [(1 - \delta_i') P_i + \delta_i' I]$ in $A_j$.

To find this for a given $\{\delta_i'\}$, we simply need to find which $\{\delta_i\}$ satisfying $\sum_i \delta_i = j$ also satisfy "$\forall i, \delta_i' = 1 \Rightarrow \delta_i = 1$." (which is just the negation of the property that leads to the coefficient being 0). We first start by letting $k = \sum_i \delta_i'$. All valid $\{\delta_i\}$ satisfy $\delta_i' = 1 \Rightarrow \delta_i = 1$, so the $i$ where $\delta_i' = 1$ forces $\delta_i = 1$. The final condition to satisfy is the sum. The forced positions sum to $k$, so it remains that the of the free positions, their total must sum to $j - k$. That is, out of the $(K - 1) - k$ free positions, exactly $j - k$ of them must be 1. Clearly, this means there are $\binom{(K-1)-k}{j-k}$ $\{\delta_i\}$'s with a non-zero $a_{\delta_1', \cdots, \delta_{K-1}'}$. Furthermore, $a_{\delta_1', \cdots, \delta_{K-1}'} = (-1)^{j-k}$ for each. Therefore, we can combine all of these together to get

$$A_j = \sum_{\delta_1', \delta_2', \cdots, \delta_{K-1}' | \sum_i \delta_i' \leq j} (-1)^{j - \sum_i \delta_i'} \binom{n - \sum_i \delta_i'}{j - \sum_i \delta_i'} H \prod_{1 \leq i \leq K-1} [(1 - \delta_i') P_i + \delta_i' I]$$

$$= \sum_{k=0}^{j} \sum_{\delta_1', \delta_2', \cdots, \delta_{K-1}' | \sum_i \delta_i' = k} (-1)^{j-k} \binom{(K-1)-k}{j-k} H \prod_{1 \leq i \leq K-1} [(1 - \delta_i') P_i + \delta_i' I]$$

$$= \sum_{k=0}^{j} (-1)^{j-k} \binom{(K-1)-k}{j-k} \sum_{\delta_1', \delta_2', \cdots, \delta_{K-1}' | \sum_i \delta_i' = k} H \prod_{1 \leq i \leq K-1} [(1 - \delta_i') P_i + \delta_i' I]$$

$$= \sum_{k=0}^{j} (-1)^{j-k} \binom{(K-1)-k}{j-k} B_k$$

$\square$

## M Disconnectedness of 2-D Orthogonal Convolutions

**Theorem 2** (Connected Components of 2-D Orthogonal Convolution with $K = 2$). *2-D orthogonal convolution space with a kernel size of $2 \times 2$ has at least $2(n+1)^2$ connected components, where $n$ is the number of channels.*

*Proof.* Consider a 2-D convolutional kernel $A$:

$$A = \begin{pmatrix} A_1 & A_2 \\ A_3 & A_4 \end{pmatrix}.$$

The orthogonality constraint implies that

$$A_1 A_4^T = 0$$
$$A_2 A_3^T = 0$$
$$A_1 A_2^T + A_3 A_4^T = 0$$
$$A_1 A_3^T + A_2 A_4^T = 0$$
$$\sum_i A_i A_i^T = I,$$

which yields

$$(A_1 + A_2)(A_3 + A_4)^T = 0$$
$$(A_1 + A_3)(A_2 + A_4)^T = 0$$
$$(A_1 + A_2 + A_3 + A_4)(A_1 + A_2 + A_3 + A_4)^T = I$$

It is clear to see that $A_1 + A_2 + A_3 + A_4$ is orthogonal; hence, we can always find an orthogonal matrix $H = (A_1 + A_2 + A_3 + A_4)^T$ such that

$$H(A_1 + A_2 + A_3 + A_4) = I$$

We can apply the orthogonal matrix to each $A_i$ with the same set of constraints:

$$\tilde{A}_i = H A_i$$

Then, we have

$$(\tilde{A}_1 + \tilde{A}_2)(\tilde{A}_3 + \tilde{A}_4)^T = 0$$
$$(\tilde{A}_1 + \tilde{A}_3)(\tilde{A}_2 + \tilde{A}_4)^T = 0$$
$$\tilde{A}_1 + \tilde{A}_2 + \tilde{A}_3 + \tilde{A}_4 = I$$

Let $P = \tilde{A}_1 + \tilde{A}_2, Q = \tilde{A}_1 + \tilde{A}_3$. We then have

$$P(I - P)^T = 0$$
$$Q(I - Q)^T = 0$$

which is equivalent of saying that $P, Q \in \mathbb{P}(n)$.

We first prove that there always exists an orthogonal convolution with arbitrary symmetric projectors $P$ and $Q$ by carefully choosing $A_i$'s:

Let $P$ and $Q$ be the symmetric projectors and $H$ be the orthogonal matrix in BCOP algorithm for $2 \times 2$ orthogonal convolution, then we have

$$A = H\square \begin{pmatrix} \tilde{A}_1 & \tilde{A}_2 \\ \tilde{A}_3 & \tilde{A}_4 \end{pmatrix} = H\square \begin{pmatrix} PQ & P(I-Q) \\ (I-P)Q & (I-P)(I-Q) \end{pmatrix}$$

with all the conditions above satisfied $P = \tilde{A}_1 + \tilde{A}_2, Q = \tilde{A}_1 + \tilde{A}_3$. Thus, we can always obtain an orthogonal convolution with arbitrary $P$ and $Q$.

Since the space of symmetric projectors is separated by rank, we can conclude that the space of $\tilde{A}_1 + \tilde{A}_2$ is disconnected and have $n + 1$ connected components ($n$ is the size of the matrices). We denote the space of special orthogonal convolution that has $\mathrm{rank}(\tilde{A}_1 + \tilde{A}_2) = p$, $\mathcal{X}_p$. Due to disconnectedness of symmetric projector, $\mathcal{X}_p \cup \mathcal{X}_{p'}$ is path-disconnected for any $p \neq p'$. Similarly, we can denote the space of orthogonal convolution that has $\mathrm{rank}(\tilde{A}_1 + \tilde{A}_3) = q$, $\mathcal{Y}_q$. $\mathcal{Y}_q$ has similar disconnectedness condition. We can define the intersection of $\mathcal{X}_p$ and $\mathcal{Y}_q$ as $\mathcal{S}_{p,q} = \mathcal{X}_p \cap \mathcal{Y}_q$ for all $p, q$. Previously, we proved that there exists orthogonal convolution for any $P$ or $Q$, so $\mathcal{S}_{p,q} \neq \varnothing$ for all $p, q$. From the disconnectedness of each of $\mathcal{X}$'s and $\mathcal{Y}$'s, we can conclude that $\mathcal{S}_{p,q}$ is disconnected from $\mathcal{S}_{p',q'}$ for any $p, q, p', q'$ if $(p, q) \neq (p', q')$.

Up to this point, we have identified $(n + 1)^2$ disjoint components of $\begin{pmatrix} \tilde{A}_1 & \tilde{A}_2 \\ \tilde{A}_3 & \tilde{A}_4 \end{pmatrix}$ (the number is induced by the combintorial selection of $p$ and $q$ from $\{0, 1, \cdots, n\}$). Combining this result with the two connected components in orthogonal matrix $H$, we can conclude that the 2-D $2 \times 2$ orthogonal convolution has at least $2(n + 1)^2$ connected components. $\qquad\square$

## N  Doubling the Channel Size Addresses BCOP Disconnectedness Issues

**Theorem 3** (BCOP Construction with Auxiliary Dimension). *For any convolution $C = \mathcal{W}(H, P_{1:K-1}, Q_{1:K-1})$ with input and output channels $n$ and $P_i, Q_i \in \mathbb{P}(n)$, there exists a convolution $C' = \mathcal{W}(H', P'_{1:K-1}, Q'_{1:K-1})$ with input and output channels $2n$ constructed from only $n$-rank projectors ($P'_i, Q'_i \in \mathbb{P}(2n, n)$) such that $C'(\mathbf{x})_{1:n} = C(\mathbf{x}_{1:n})$. That is, the first $n$ channels of the output is the same with respect to the first $n$ channels of the input under both convolutions.*

*Proof.* **(Theorem 3)** We will start by defining two functions $f, f'$ to be "effectively equivalent" if $f'(\mathbf{x})_{1:n} = f(\mathbf{x}_{1:n})$. Notice that the input and output are image tensors and $1 : n$ is across the channel dimension. It is clear to see that if $f, f'$ are effectively equivalent and so are $g, g'$, the $g \circ f$ is effectively equivalent to $g' \circ f'$. The same holds for $f + g$ and $f' + g'$.

Now we will construct the parameters of $C'$ from the parameters of $C$. For any $n \times n$ projector $P$ used as a parameter for $C$, define $P'$ as,

$$P' = \begin{pmatrix} P & 0 \\ 0 & S_k \end{pmatrix}$$

where $k$ is the rank of $P$ and $S_k$ is a diagonal matrix with the first $n - k$ entries as 1, and the rest as 0. Notice that $\text{rank}(P') = \text{rank}(P) + \text{rank}(S_k) = k + (n - k) = n$, which independent of the rank of $P$. Finally, replace the orthogonal parameter $H$ for $C$ with

$$H' = \begin{pmatrix} H & 0 \\ 0 & I \end{pmatrix}$$

This covers all the parameters in BCOP, so we can now construct $C'$ from $C$ using only $n$-rank projectors. It is easy to see that each projector $P'$ and $H'$ were constructed such that they are effectively equivalent to their counterpart in $C$. Now notice the convolution $C'$ is computed as

$$C' = H' \square \begin{bmatrix} P'_1 & I - P'_1 \end{bmatrix} \square \begin{bmatrix} Q'_1 \\ I - Q'_1 \end{bmatrix} \square \cdots \square \begin{bmatrix} P'_{K-1} & I - P'_{K-1} \end{bmatrix} \square \begin{bmatrix} Q'_{K-1} \\ I - Q'_{K-1} \end{bmatrix}$$

and by the properties of block convolution, this equivalent to applying each of these functions from right to left to an input image tensor. Thus, by the composition rule for effective equivalence, if all of these functions are equivalent to their $n$ channel counterpart, then $C'$ is effectively equivalent to $C$. Each of these size 2 convolutions computes each output position as a function of two input positions, By first applying a projector to each ($P'$ is applied to one and $I - P'$ is applied to another) and then summing. By the properties of effective equivalence, this means that the application of each size 2 convolution is effectively equivalent to its corresponding convolution in $C$. Multiplying by the orthogonal matrix $H'$ is also effectively equivalent to $H$. Therefore, each function comprising $C'$ is effectively equivalent to its counterpart in $C$, so $C'$ is is effectively equivalent to $C$. $\square$

The theorem above shows that we can represent any $n$-dimensional BCOP convolution with a $2n$-dimensional BCOP convolution that only has rank $n$ projectors, which is a connected space. This allows us to circumvent the disconnectedness issues that arise in our analysis in Appendix M by doubling the size of all symmetric projectors by a factor of 2, and set them to be exactly half of the full rank.

However, in order to use this, the input would need $n$ "dummy" dimensions so that it can pass through a $2n$-dimensional convolution. This can simply be set up in the initial convolution of a BCOP-parameterized network. The initial convolution would initially have an orthogonal $n \times m$ matrix $H$ to upsample from the network input's $m$ channel size to the desired $n$ channel size ($m$ is typically significantly less than $n$). If we expand $H$ by simply adding $n$ rows of zeros underneath, then we will have a $2n \times m$ matrix which preserves the first $n$ channels of the output while also maintaining orthogonality of $H$. The projectors in this upsampling layer are $n \times n$, so we can simply enlarge these in the same way as in the above theorem to get the desired result from this layer. Finally, we need to address the transition from the convolution layers to the fully-connected layer. This is also straightforward as we can add $n$ columns of 0s to the first fully-connected layer's weight matrix. In this way, the dummy dimensions will have no impact on the network's output. Thus, any network using BCOP convolutions can be equivalently represented in a single connected component of networks using BCOP convolutions with double the channel size. Therefore, this presents a way to circumvent the disconnectedness issue.

## O  Incompleteness of 2-D Convolution Parameterization

Xiao et al. [24] extends the 1-D orthogonal convolution construction algorithm presented in Kautsky and Turcajová [15] to construct 2-D orthogonal kernel as follows:

$$A = H\square \begin{bmatrix} P_1 \\ I - P_1 \end{bmatrix} \square [Q_1 \quad I - Q_1] \square \cdots$$
$$\cdots \square \begin{bmatrix} P_{K-1} \\ I - P_{K-1} \end{bmatrix} \square [Q_{K-1} \quad I - Q_{K-1}] \tag{9}$$

where $X\square Y = \left[\sum_{i',j'} X_{i',j'} Y_{i-i',j-j'}\right]_{i,j}$ with the out-of-range matrices all zero, $H$ is an orthogonal matrix, and $P_1, \cdots P_{K-1}$ and $Q_1, \cdots Q_{K-1}$ are symmetric projectors. However, unlike in the 1-D case, the 2-D orthogonal kernels constructed from this algorithm doesn't cover the entire orthogonal kernel space.

To illustrate this, we first consider a $2 \times 2$ convolution kernel $A = \begin{bmatrix} A_1 & A_2 \\ A_3 & A_4 \end{bmatrix}$ with its equivalent transformation matrix over a $3 \times 3$:

$$\begin{bmatrix} A_1 & A_2 & 0 & A_3 & A_4 & 0 & 0 & 0 & 0 \\ 0 & A_1 & A_2 & 0 & A_3 & A_4 & 0 & 0 & 0 \\ A_2 & 0 & A_1 & A_4 & 0 & A_3 & 0 & 0 & 0 \\ 0 & 0 & 0 & A_1 & A_2 & 0 & A_3 & A_4 & 0 \\ 0 & 0 & 0 & 0 & A_1 & A_2 & 0 & A_3 & A_4 \\ 0 & 0 & 0 & A_2 & 0 & A_1 & A_4 & 0 & A_3 \\ A_3 & A_4 & 0 & 0 & 0 & 0 & A_1 & A_2 & 0 \\ 0 & A_3 & A_4 & 0 & 0 & 0 & 0 & A_1 & A_2 \\ A_4 & 0 & A_3 & 0 & 0 & 0 & A_2 & 0 & A_1 \end{bmatrix}$$

The equivalent conditions for the convolution kernel to be orthogonal can be summarized as follows (inner products of any pairs of distinct rows need to be 0):

$$A_1 A_4^T = 0 \text{ (Inner product of Row 1 and Row 9)}$$
$$A_2 A_3^T = 0 \text{ (Inner product of Row 1 and Row 8)}$$
$$A_1 A_2^T + A_3 A_4^T = 0 \text{ (Inner product of Row 1 and Row 2)}$$
$$A_1 A_3^T + A_2 A_4^T = 0 \text{ (Inner product of Row 1 and Row 7)}$$
$$\sum_i A_i A_i^T = I \text{ (Self inner product of any row)}$$

However, notice that since BCOP $2 \times 2$ convolution is of the form,

$$A = H\square \begin{bmatrix} P \\ I - P \end{bmatrix} \square [Q \quad I - Q] = \square \begin{bmatrix} HPQ & HP(I-Q) \\ H(I-P)Q & H(I-P)(I-Q) \end{bmatrix}$$

It is clear we will always have an additional constraint of $A_1 A_2^T = (HPQ)(HP(I-Q))^T = 0$. However, we can find an orthogonal matrix that does not satisfy this condition by defining the $A_1$ to $A_4$ as:

$$A_1 = \frac{1}{2} \begin{bmatrix} 1 & 0 \\ -1 & 0 \end{bmatrix}$$

$$A_2 = \frac{1}{2} \begin{bmatrix} 1 & 0 \\ 1 & 0 \end{bmatrix}$$

$$A_3 = \frac{1}{2} \begin{bmatrix} 0 & -1 \\ 0 & 1 \end{bmatrix}$$

$$A_4 = \frac{1}{2} \begin{bmatrix} 0 & 1 \\ 0 & 1 \end{bmatrix}$$

However, while this does show that BCOP is incomplete, the counterexample is fairly uninteresting. In fact, it can be represented as

$$A = [P \quad I - P] \square \begin{bmatrix} Q \\ I - Q \end{bmatrix}, \text{ where } P = \begin{bmatrix} 1 & 0 \\ 0 & 0 \end{bmatrix}, Q = \frac{1}{2} \begin{bmatrix} 1 & -1 \\ -1 & 1 \end{bmatrix}$$

which is simply a re-ordering of the 1-D convolutions in the BCOP definition. Furthermore, it can even be trivially represented by composing two BCOP convolutions together. It is still an open question as to whether or not allowing re-ordering of the 1-D convolution components of BCOP will be able to represent all 2-D orthogonal convolutions (while re-ordering 1-D convolutions is not something easily achievable in neural network architectures, composing arbitrarily many BCOP convolutions can represent any re-ordering of 1-D convolutions).

## P  Certifying Provable Robustness for Lipschitz Network

To certify provable robustness of a Lipschitz network, we first define the margin of a prediction for a data point $x$,

$$\mathcal{M}_f(\mathbf{x}) = \max(0, y_t - \max_{i \neq t} y_i)$$

where $\mathbf{y} = [y_1, y_2, \cdots]$ is the predicted logits from the model on data point $\mathbf{x}$ and $y_t$ is the correct logit ($\mathbf{x}$ belongs to $t^{\text{th}}$ class). Following the results from Anil et al. [2], Tsuzuku et al. [23], we derive the sufficient condition for a data point to be provably robust to perturbation-based adversarial examples in the general case:

**Theorem 7** (Adversarial Perturbation Robustness Condition under $L_p$ Norm). *If $2^{\frac{p-1}{p}} l\epsilon < \mathcal{M}_f(\mathbf{x})$, where $f$ is an $l$-Lipschitz under the $L_p$ norm, then $\mathbf{x}$ is robust to any input perturbation $\Delta\mathbf{x}$ with $||\Delta\mathbf{x}||_p \leq \epsilon$.*

*Proof.* **(Theorem 7)**
Let the function that represents the network to be $f$, $\mathbf{x}$ be some data point, and $\mathbf{y} = f(\mathbf{x})$.

We see that it is enough to consider when $\mathbf{x}$ is correctly classified. If it were misclassified, $\mathcal{M}_f(\mathbf{x}) = 0$, so $2^{\frac{p-1}{p}} l\epsilon < \mathcal{M}_f(\mathbf{x})$ can never hold. Since $\mathbf{x}$ is correctly classified, $\mathcal{M}_f(\mathbf{x}) \leq y_t - y_i$ for all $i \neq t$ where $t$ is the correct class.

Now suppose there is a $\Delta\mathbf{x}$ such that $||\Delta\mathbf{x}||_p \leq \epsilon$ and $\mathbf{x}' = \mathbf{x} + \Delta\mathbf{x}$ is incorrectly classified. Then, $y_t' \leq y_w'$ for some $w$ where $\mathbf{y}' = f(\mathbf{x}')$. We also denote the perturbation in the $i^{\text{th}}$ dimension to be $\Delta y_i = y_i' - y_i$.

Since $f$ is $L$-Lipschitz, we can bound the norm of the change of output as follows:

$$\left( \sum_i |\Delta y_i|^p \right)^{\frac{1}{p}} = ||f(\mathbf{x} + \Delta\mathbf{x}) - f(\mathbf{x})||_p \leq l||\Delta\mathbf{x}||_p \leq l\epsilon$$

Then we have

$$|\Delta y_t|^p + |\Delta y_w|^p \leq \sum_i |\Delta y_i|^p \leq (l\epsilon)^p$$

Now consider a polynomial $g(r) = r^p + (\Delta y_w - \Delta y_t - r)^p$. By analyzing derivatives, this has a global minimum at $r = \frac{\Delta y_w - \Delta y_t}{2}$ using the fact that $\Delta y_w \geq \Delta y_t$. This yields,

$$\frac{|\Delta y_w - \Delta y_t|^p}{2^{p-1}} = g\left( \frac{\Delta y_w - \Delta y_t}{2} \right) \leq g(-\Delta y_t) = (-\Delta y_t)^p + (\Delta y_w)^p \leq |\Delta y_w|^p + |\Delta y_t|^p \leq (l\epsilon)^p$$

This means that

$$|\Delta y_w - \Delta y_t| \leq 2^{\frac{p-1}{p}} l\epsilon \Rightarrow -2^{\frac{p-1}{p}} l\epsilon \leq \Delta y_t - \Delta y_w$$

Substituting this bound along with the inequality $y_t - y_w \geq \mathcal{M}_f(\mathbf{x})$ yields

$$0 \geq y_t' - y_w' = y_t - y_w + (\Delta y_t - \Delta y_w) \geq \mathcal{M}_f(\mathbf{x}) - 2^{\frac{p-1}{p}}(l\epsilon).$$

Therefore, $\mathcal{M}_f(\mathbf{x}) \leq 2^{\frac{p-1}{p}}(l\epsilon)$.

Thus, by contrapositive $\mathcal{M}_f(\mathbf{x}) > 2^{\frac{p-1}{p}}(L\epsilon)$ implies $y_t' - y_w' > 0$. $\qquad\square$

## Footnotes

[1]Convolving a transposed kernel is equivalent to applying a transposed linear transformation