[Reviews · NeurIPS 2019]

Reviewer 1



Originality. This work is a followup of Anil et al. 2019 and extends their approach to the case of convolutional layers. As so, this is incremental work, although some hurdles have to be overcome in order to extend the results in this new setting. The BCOP parametrization of gradient-norm preserving convolutional layers seems a novel application of previous work (Xiao et al. 2018). The disconectedness result for 1-D convolutions adds to the understanding of the difficulty of the problem. Quality. The proposed algorithms are technically and theoretically sound. However I find some parts are lacking. (1) There is no mention or discussion about how could the method adapt to other metrics rather than the L2 metric, which makes the method somewhat limited. (2) There are some claims that are either incorrect or have no reference e.g. (2a) section 2.2 second paragraph: "one can show that the norm of the gradient backpropagating through a 1-Lipschitz...", this statement is not clear at all and moreover does not point out to any reference. (2b) 3.1.1. second paragraph: "this is a valid projection under the matrix 2-norm but not the Frobenius norm", this is a reference to a paper by Gouk et al. 2018. where a matrix is scaled by its matrix 2-norm, however in my understanding this is NOT the projection onto the 1-ball in matrix 2-norm, rather one has to do singular value clipping and that would be the projection both w.r.t. 2-norm and Frobenius norm. (Edit: I stand corrected as the rescaling is indeed a valid projection under the 2 operator norm) (2c) same paragraph: "... is not guaranteed to converge to the correct solution" why? reference? (2d) next paragraph: "... permits Euclidean steepest descent", I might be missing something but I am not sure what is meant by this. (2e) 3.1.1. first paragraph: "The Lipschitz constant of the convolution operator is bounded by *a constant factor* of the spectral norm of the kernel reshaped into a matrix", perhaps I am missing something but can the authors comment on where does this "constant factor" come from? I would assume that the spectral norm of the kernel reshaped into a matrix is exactly the Lipschitz constant of the convolution, given that the convolution operator is equivalent to the matrix form (they are the same function). (2f) section 4.4. first paragraph "ensuring gradient norm preservation is critical for obtaining tighter lower bounds on the Wasserstein distance", as I understand this is not critical, in the sense that there might be better methods to approximate 1-Lipschitz functions that do not rely on gradient norm preservation, If there is some negative result stating that this is not possible without gradient norm preservation, there should be a clear reference. (3) The results from table 1 shows marginal improvements over existing method, for example 51.47 vs 50.00 or 49.37 vs 48.07. Without any sort of confidence intervals or repeated trials it is too difficult to assess how much of an improvement this is. The same for the other tables. Clarity. The general idea of the paper can be grasped on one reading, however there are many details that are confusing and that need improvement (1) Throughout the paper there are many references to a term "Lipschitz networks" What do you mean by this? as long as the activations are Lipschitz continuous (most of them are) then any network is Lipschitz continuous. Maybe you want to say "networks with a small Lipschitz constant...". Again in the second paragraph you say "... ensure that each piece of the computation is Lipschitz", what do you mean by piece? I guess you mean layers or something similar, in this case again any linear layer with Lipschitz activation is Lipschitz, perhaps you mean to say "ensure that each layer has a small Lipschitz constant". This makes the claims confusing. (2) I feel there are some functions that are not properly introduced and one has to make an effort to understand. The notation Bjorck(R) could be introduced better, and the notation SymmetricProjector(M) is not introduced at all before presenting BCOP. (3) When the experimental section starts there is no clarity about how the networks were trained. I think it is only explicitely stated later in the last page "we implicitely enforce the Lipschitz constant of the network to be exactly 1" this should be made clear before the experiments sections. Significance: I think the extension to the convolutional case is important as it is a widely used type of layer and the results of the previous work by Anil et al. 2019 seem promising. However the experimental part needs some "technical" work to better assess the improvements that one can see in practice.

Reviewer 2



This paper studies the block convolutional orthogonal representation for convolutional neural networks in presence of Lipschitz constraints. During training of the network gradient norm preservation is utilized to combat gradient attenuation. This simple combination of the two main ideas outlined above is shown to be useful in two settings, namely adversarial training and computing the Wasserstein distance using Kantorovic duality. Admittedly, I might have missed important pieces of the paper, but I regret to say that the paper is limited in terms of its novelty. As is, the paper feels like a combination of two existing ideas from Anil et al. and Xiao et al. Having a broader set of experiments could have made up for the limited novelty of the paper, I would argue. For example, the Wasserstein computation is a crucial step in performing Wasserstein GAN, so I was really curious to see if the benefits ultimately transfers to better generative modeling. On the same note, another application could have been learning stochastic models of the world in reinforcement learning, where it is is important to compute models with low Wasserstein errors.

Reviewer 3



[Originality] Although the authors point out that there exist papers on Lipschitz convolutional networks, they are not expressive enough to perform well. And analyzing the Lipschitz property of CNNs is harder than that of fully-connected nets. Therefore, I consider this paper a novel and original extension of previous work for Lipschitz networks (on fully-connected networks). [Quality] The proposed method (BCOP) is technically sound. The visualizations of singular values of weights of trained network layers clearly indicate that BCOP works as predicted in theory. The authors show empirical results on both adversarial robustness and estimating Wasserstein distance between data distribution and generated image distribution of GAN. I believe these results justify the effectiveness of BCOP. [Clarity] The paper is well-written and enough implementation details are provided for readers to implement BCOP. Moreover, the authors explain the derivation in depth in the appendix. [Significance] As mentioned in originality section, this paper is the first paper that does not place tight constraints on expressiveness of CNNs to constrain the Lipschitz constant while keeping gradient not vanishing. As Lipschitz network has been shown important in many tasks like GAN, adversarial robustness and invertible nets, this paper definitely makes a major contribution to the community. In addition, this paper provides theoretical insights for others to work on Lipschitz properties of neural nets.

[Author Response · NeurIPS 2019]

| Dataset | | | OSSN | RKO | SVCM | BCOP |
|---|---|---|---|---|---|---|
| **CIFAR10** $(\epsilon = 36/255)$ | **Small** | Clean | $61.83 \pm 0.86$ | $61.75 \pm 0.57$ | $61.88 \pm 0.52$ | $\mathbf{64.25} \pm 0.39$ |
| | | Robust | $47.77 \pm 0.74$ | $47.39 \pm 0.44$ | $47.17 \pm 0.41$ | $\mathbf{49.95} \pm 0.17$ |
| | **Large** | Clean | $68.28 \pm 0.51$ | $69.47 \pm 0.24$ | $69.44 \pm 0.26$ | $\mathbf{72.01} \pm 0.24$ |
| | | Robust | $54.26 \pm 0.40$ | $55.41 \pm 0.21$ | $53.57 \pm 0.18$ | $\mathbf{58.26} \pm 0.17$ |

We thank all the reviewers for their thorough feedback. We have updated all tables to report results from 5 repeated
trials. Our reported improvements are consistent throughout. We would like to re-emphasize our main contributions
here: **(1)** To the best of our knowledge, we are the first to reveal the disconnectedness of the space of orthogonal
convolutions. We believe our analysis demonstrates this space is unexpectedly complicated and inherently difficult to
optimize over. **(2)** We analyze and identify the shortcomings of existing methods of enforcing Lipschitz-constrained
convolutions. In particular, we find gradient attenuation to be a common problem among many of these methods and
propose using orthogonal convolutions to circumvent this. **(3)** We adapt Xiao et al. [40]*'s orthogonal convolution
initialization procedure to be used for optimizing over the orthogonal convolution space. Our parameterization alleviates
the issues of the disconnected orthogonal convolution space that arose in our analysis. We verified its effectiveness on
adversarial robustness and Wasserstein distance estimation tasks over the pre-existing methods.

**Reviewer 1:** All empirical results now include error bars (see example table, top). We observed statistical significance
throughout. We discuss other points below.

*Quality - (1)* Our BCOP parameterization lies in the space of orthogonal convolutions, which is 1-Lipschitz only under
the $L_2$ metric. We will make clear that we focus on Lipschitz convolutional networks with the $L_2$ metric only.

*Quality - (2a)* To clarify, the statement was trying to demonstrate a relationship between the gradient norm before and
after back-propagating through a 1-Lipschitz function. To be precise, let $\mathbf{y} = f(\mathbf{x})$ for some 1-Lipschitz $f$, and $\mathcal{L}(\mathbf{y})$
be a loss function. We have $\left\| \frac{\partial \mathcal{L}}{\partial \mathbf{x}} \right\|_2 = \left\| \frac{\partial \mathcal{L}}{\partial \mathbf{y}} \frac{\partial \mathbf{y}}{\partial \mathbf{x}} \right\|_2 \leq \left\| \frac{\partial \mathcal{L}}{\partial \mathbf{y}} \right\|_2 \left\| \frac{\partial \mathbf{y}}{\partial \mathbf{x}} \right\|_2 \leq \mathrm{Lip}(f) \left\| \frac{\partial \mathcal{L}}{\partial \mathbf{y}} \right\|_2 = \left\| \frac{\partial \mathcal{L}}{\partial \mathbf{y}} \right\|_2$, where $\left\| \frac{\partial \mathcal{L}}{\partial \mathbf{x}} \right\|_2$ and
$\left\| \frac{\partial \mathcal{L}}{\partial \mathbf{y}} \right\|_2$ are the input and output gradient norm correspondingly, and $\mathrm{Lip}(f)$ is the Lipschitz constant of the function $f$.
We will adjust the phrasing of the statement in the paper to include this detailed explanation.

*Quality - (2b, 2c, 2d)* Gouk et al. [15]* [1] write that OSSN will "project it back to the closest matrix in the feasible
set measured by the matrix distance metric induced by taking the operator norm". One can prove that OSSN is a
valid projection under 2-norm but not Frobenius norm (while SV clipping is a valid projection for both norms as R1
suggested). Because OSSN uses a different norm for the steepest descent direction and the projection step, it's not
guaranteed to converge; we give a counterexample in Section A of the supplemental material.

*Quality - (2e)* By "reshaping a kernel into a matrix", we were referring to **flattening** a 2-D convolution kernel tensor of
shape $(c_o, c_i, k, k)$ into a $c_o \times c_i k^2$ matrix, where $c_i, c_o, k$ are input channel size, output channel size and kernel size,
respectively; whereas, the **matrix form** of the convolution operator is a $hwc_o \times hwc_i$ matrix ($h, w$ are the input/output
spatial dimensions). Tsuzuku et al. [36]* has shown a constant factor of the spectral norm of the **"reshaped/flattened**
**kernel"** bounds the Lipschitz constant of the convolution operator, arising from the repetitions of each convolution
kernel tensor element in the **matrix form** due to overlapping convolution windows.

*Quality - (2f)* The optimal dual function must have gradient norm 1 almost everywhere on the support (see Corollary 1 in
Gemicic et al. [41]), which can be achieved by gradient norm preservation throughout the network. However, we did not
mean to imply that limiting the function space we are optimizing over to be gradient norm preserving is theoretically the
best way to estimate Wasserstein distance. We will adjust the writing and supply the additional references accordingly.

*Clarity* As pointed out by the reviewer, "Lipschitz network" indeed refers to a network with a specified Lipschitz
constant that is enforced tightly. This will be clarified. We will re-organize method and experiment sections to clarify
notations and key experimental details.

**Reviewer 2** expressed concerns over novelty of this work. As discussed above, we do not simply combine methods
from Xiao et al. [40]* and Anil et al. [1]*. Xiao et al. [40]*'s algorithm is used for initializing orthogonal convolutions
while we need to parameterize the orthogonal convolution space to be optimized over. Moreover, our theoretical analysis
enables our BCOP parameterization to be configured to maximize the expressiveness of the orthogonal convolution.

**Reviewer 3** inquired about run-time comparison of our Lipschitz convolutional network against standard non-Lipschitz
convolutional network. During the training of the "large" architecture described in the paper, [2] the BCOP-parameterized
network takes 0.138 seconds per training iteration while a standard non-Lipschitz network with the same architecture
only takes 0.041 seconds per training iteration. As for the other Lipschitz methods, RKO takes 0.120 seconds and
OSSN (with one power iteration) takes 0.113 seconds. We will report these values in the paper.

**Additional Reference:** [41] Mevlana Gemici, Zeynep Akata, and Max Welling. Primal-dual Wasserstein GAN. 2018.

## Footnotes

[1] The starred references are from the original paper. Any additional references are provided below.

[2] Training speed benchmark setup: CIFAR10, NVIDIA P100, batch size of 128.


[Meta-Review · NeurIPS 2019]

This paper describes a novel parametrization method for gradient-norm-preserving Lipschitz convolutional network. It is an extension of Anil et al. 2019 to the case the CNN. The paper is well-written and the technical depth qualifies NeurIPS. The proposed method seems to be a combination of exisiting ideas, so the novelty is a bit limited (but still ok for NeurIPS). The authors are encouraged to consider the reviews seriously to further improve the paper quality.